# Extremely Warm European Summers preceded by Sub-Decadal North Atlantic Ocean Heat Accumulation

Lara Wallberg[1,2], Laura Suarez-Gutierrez[1,3,4], Daniela Matei[1], and Wolfgang A. Müller[1]

[1]Max Planck Institute for Meteorology, Hamburg, Germany
[2]International Max Planck Research School on Earth System Modelling (IMPRS-ESM), Hamburg, Germany
[3]Institute of Atmospheric and Climate Science, ETH Zürich, Zurich, Switzerland
[4]Institut Pierre-Simon Laplace, CNRS, Paris, France

**Correspondence:** Lara Wallberg (lara.wallberg@mpimet.mpg.de)

**Abstract.** The internal variability of European summer temperatures has been linked to various mechanisms on seasonal to sub- and multi-decadal timescales. We find that sub-decadal time scales dominate summer temperature variability over large parts of the continent and determine a mechanisms controlling extremely warm summers on sub-decadal time scales. We show that the sub-decadal warm phases of bandpass-filtered European summer temperatures, hereinafter referred to as extremely warm European summers, are related to a strengthening of the North Atlantic ocean subtropical gyre, an increase of meridional heat transport, and an accumulation of ocean heat content in the North Atlantic several years prior to the extreme summer. This ocean warming affects the ocean-atmosphere heat fluxes, leading to a weakening and northward displacement of the jet stream and increased probability of occurrence of high pressure systems over Scandinavia. Thus, our findings link the occurrence of extremely warm European summers to the accumulation of heat in the North Atlantic Ocean, and provide the potential to improve the predictability of extremely warm summers several years ahead which is of great societal interest.

## 1 Introduction

Extremely warm European summers have an increasingly large societal impact. Extreme temperatures can lead to severe health problems and are thus associated with an increased mortality (Gasparrini et al., 2015; Vicedo-Cabrera et al., 2021). Furthermore, heat extremes can also lead to economic impacts, such as crop failure and water shortages (Ribeiro et al., 2020), along with political challenges, including climate-induced migration and the need for effective crisis management (Ceglar et al., 2019). European summers will become more extreme in a warming climate due to rising mean temperatures (Seneviratne et al., 2021) and also due to an increase in internal temperature variability (Schär et al., 2004; Fischer et al., 2012; Suarez-Gutierrez et al., 2020a). Moreover, when such extreme summers occur repeatedly year after year, they become even more threatening to the already vulnerable socioeconomic and ecological resilience of the region (Ruiter et al., 2020; Callahan and Mankin, 2022).

On time scales of days to several weeks, the main drivers of extreme heat are soil moisture deficits and moisture-temperature feedbacks (Seneviratne et al., 2006; Fischer and Schär, 2008; Vogel et al., 2017; Suarez-Gutierrez et al., 2020a; Röthlisberger and Papritz, 2023) and large-scale atmospheric patterns such as atmospheric blocking and the North Atlantic Oscillation (Meehl and Tebaldi, 2004; Horton et al., 2015; Li et al., 2020; Suarez-Gutierrez et al., 2020a). However, these short-term drivers of

extreme temperatures could be influenced and conditioned by mechanisms on longer time scales. Long memory mechanisms such as ocean heat inertia, i.e., the capacity to store heat and delay its transfer and release, have been found to influence mean summer temperature variability (Saeed et al., 2013; Ghosh et al., 2016; Borchert et al., 2019). Examples for these long-term mechanisms influencing European temperatures are the Atlantic multi-decadal variability (AMV; Boer et al., 2016; Gao et al., 2019; Qasmi et al., 2021; Ruprich-Robert et al., 2021) or the El-Nino Southern Oscillation (ENSO; Martija-Díez et al., 2021). The variability in the North Atlantic region has been shown to include a fully coupled atmosphere-ocean cycle with a period of about 7-10 years shown for different ocean-related quantities, such as ocean heat content and barotropic stream function (Reintges et al., 2016; Martin et al., 2019). In fact, these processes have a significant impact on European summer temperatures as demonstrated by Müller et al. (2020). However, the assessment of drivers for extreme temperatures on such long-term timescales is currently limited (Simpson et al., 2018; Wu et al., 2019), and their relevance for extreme summers remains uncertain (Röthlisberger and Papritz, 2023). This research addresses this question and presents a comprehensive explanation for the occurrence of extremely warm European summers in sub-decadal warm phases, and their relation to the heat accumulation that occurs several years in advance.

Our investigation concentrates on the exceptionally warm European summers that occur in conjunction with positive sub-decadal temperature anomalies. To robustly capture the frequency and strength of such low-probability events, large sample sizes are needed. Here, we use one of the largest ensembles from a comprehensive, fully coupled Earth system model currently available, the Max-Planck-Institute Grand Ensemble with 100 ensemble members (MPI-GE; Maher et al., 2019). MPI-GE offers one of the most adequate representations of observed historical temperatures among single-model large climate models (Suarez-Gutierrez et al., 2021). MPI-GE is able to capture extreme summer temperatures (Suarez-Gutierrez et al., 2020b), including some of the most extreme European summer temperatures ever recorded (Suarez-Gutierrez et al., 2018, 2020a).

Using the MPI-GE, we examine the sub-decadal variability of extremely warm European summers and show how these summers are affected by North Atlantic Ocean heat content accumulation. We investigate whether the MPI-GE can represent sub-decadal temperature variability well, and identify where these time scales dominate over Europe and are linked to European extreme temperatures. Additionally, we identify which processes in the North Atlantic Ocean is responsible for the increase of the occurrence of extremely warm summers.

## 2 Data and Methods

### 2.1 Model Description

We use simulations from the Max Planck Institute Grand Ensemble (MPI-GE; Maher et al., 2019). These simulations are performed with MPI-ESM1.1 in the low-resolution setup (MPI-ESM-LR; Mauritsen et al., 2012; Giorgetta et al., 2013). MPI-GE consists of 100 simulations with different initial conditions and is one of the largest ensembles of a single, comprehensive, fully-coupled climate model. In the atmosphere, the MPI-ESM-LR reaches up to 0.01 hPa (about 80 km) with 47 vertical levels and a horizontal resolution of 200km at the equator. In the ocean, the MPI-ESM-LR is formulated on a C grid and orthogonal

curvilinear coordinates (Marsland et al., 2003). To circumvent grid singularities at the geographical North Pole, the northern grid pole is shifted to Greenland, leading to high resolution in the Arctic and the high-latitude sinking regions. In the ocean, the MPI-ESM-LR has 40 vertical levels and a horizontal resolution of about 1.5° on average and varies from a minimum of 12km close to Greenland to a maximum of 180km in the tropical Pacific. Here, We are using monthly data averaged to seasonal summer means over June, July, and August (JJA) from 1950 to 2022. This includes historical simulations from 1950 to 2005 and RCP4.5 scenario simulations until 2022. For time-lagged analyses up to three years prior to 1950 are analyzed.

ERA5 data including the backward extension until 1950 are used as an observational reference to validate the results of the multi-tapering with MPI-GE (Hersbach et al., 2018).

## 2.2 Analysis Methods

We linear detrended all of our data in order to exclude the influence of global warming and other external forcings. Furthermore, we use a 5-10 year bandpass filter to remove frequencies and noise outside the sub-decadal range. Therefore, we use a standard top-hat filter response function.

In order to investigate extremely warm European summers on sub-decadal timescales, hereafter referred to as extremely warm summers, we consider those JJA mean temperature anomalies in the region between 15°-35°E and 45°-60°N exceed their 90th percentile and additionally occur in a positive bandpass-filtered phase (pooled in time and ensemble (T>90th percentile and T$bandpass$ >0), 557 summers in total).

We use a cross-spectral analysis, based on a multi-taper method to analyze if and where the MPI-GE and ERA5 can represent the sub-decadal time scales (Årthun et al., 2018). This multi-taper method is a spectral analysis technique to estimate the dominant time-frequency content of time series by decomposing the data into a set of orthogonal tapers and computing a set of spectral estimates. The dominant time-frequency is then identified as the highest spectral peak or mode in the resulting spectrum, which characterize the dominant oscillatory patterns and variability of the data over time. We perform the multi-tapering for all 100 ensemble members and take the mean over all spectra for each grid point to ascertain the dominant timescale, where the dominant timescale is given by the highest significant peak (e.g. Årthun et al., 2018; Ghil et al., 2002). The significance of spectral peaks is determined by comparison with a red noise spectrum with a 95% confidence interval.

The significance of our results is tested with a bootstrap algorithm in which a reference index is computed in each grid point for 1000 randomly composed arrays of the corresponding variable (random sampling with replacement). We calculate the p-value from our 1000 bootstraps and control for the false discovery rate (equation 3 in Wilks (2016)) with a chosen control level of $\alpha_{\text{FDR}} = 0.1$.

We scale the band-pass filtered summer mean anomalies by the standard deviation of unfiltered summer (JJA) mean anomalies during extremely warm European summers to better illustrate the imprint of the sub-decadal proportion of various climate variables on the occurrence of extremely warm European summers. In detail, we first calculate anomalies of the variables with respect to their long-term averages. We then define their total summer mean variability ($\sigma_t$) as the standard deviation of the

unfiltered summer mean anomaly ($x'$) for years showing an extremely warm European summer ($t_{\text{extreme}}$):

$$\sigma_t = \sigma(x'(t_{\text{extreme}}))$$

We then divide the bandpass-filtered anomaly ($\tilde{x}'$) by the total variability $\sigma_t$ at the time of each heat extreme. Lastly, we average over all cases of extreme events ($N$) to obtain the scaled anomaly ($\widehat{x}$):

$$\widehat{x} = \frac{1}{N} \sum_{1}^{N} \frac{\tilde{x}'}{\sigma_t}$$

All calculations of the scaled anomaly are performed gridpoint-wise. The scaled anomaly simply illustrates the proportion that a sub-decadal mean change has on the occurrence of an extremely warm summers compared to the overall occurrence of extremely warm summers. The corresponding scaled anomalies are added to the respective figure captions.

## 3 Results

### 3.1 Sub-Decadal Variability and Extremely Warm European Summers

We use bandpass-filtering and a cross-spectral analysis to identify the dominant time-frequencies of European summer temperatures for each grid point in MPI-GE and ERA5 from 1950 to 2020 (Fig. 1a,b and Methods). Areas with dominant sub-decadal variations (5-10 year variations) are found in MPI-GE over Scandinavia, the British Isles, the Iberian Peninsula, Italy, and large parts of Central and Eastern Europe. ERA5 and MPI-GE show high agreement for areas with dominant sub-decadal variations especially over Eastern Europe (Fig. 1a,b). MPI-GE reveals some limitations in the representation of multi-decadal time scales (>20 years), which are dominant in ERA5 in the northern and southernmost parts of the domain. On time scales between 10 and 20 years, only a few grid points are dominant. Even fewer dominant grid points are found on time scales greater than 20 years.

Analyzing the ratio between all heat extremes and those occurring in a positive bandpass filtered phase, Central Europe stands out as the area with the highest percentages (Fig. 1c). Regions within the Iberian Peninsula, northern Scandinavia, and Russia also stand out with coinciding extremely warm summers and sub-decadal variability. In summary, sub-decadal timescales of 5-10 years are the dominant scale of variability in European mean summer temperatures, and extremely warm summers tend to occur in 5-10 year phases of abnormally warm temperatures over Europe over large parts of Central, Eastern and Southern Europe.

The overlap between a dominant sub-decadal variability and the occurrence of extremely warm European summers is strongest over Central Europe (Central Europe is defined by 15°-35°E; 45°-60°N, Fig. 1c, blue box). The temperature of this region is also dominated by the sub-decadal time scales overall (Fig. 1d), as expected from Fig. 1b, two significant peaks within the sub-decadal time scales can be found here as well. Other significant peaks could be found around two to three years and around 15 years, indicating the possible influence of other drivers and mechanisms. We investigate the behaviour of several variables that characterize the North Atlantic Ocean heat content variability during extremely warm European summers as

well as several years prior their occurrence to further understand the simultaneous occurrence and the drivers of sub-decadal variability and extremely warm European summers.

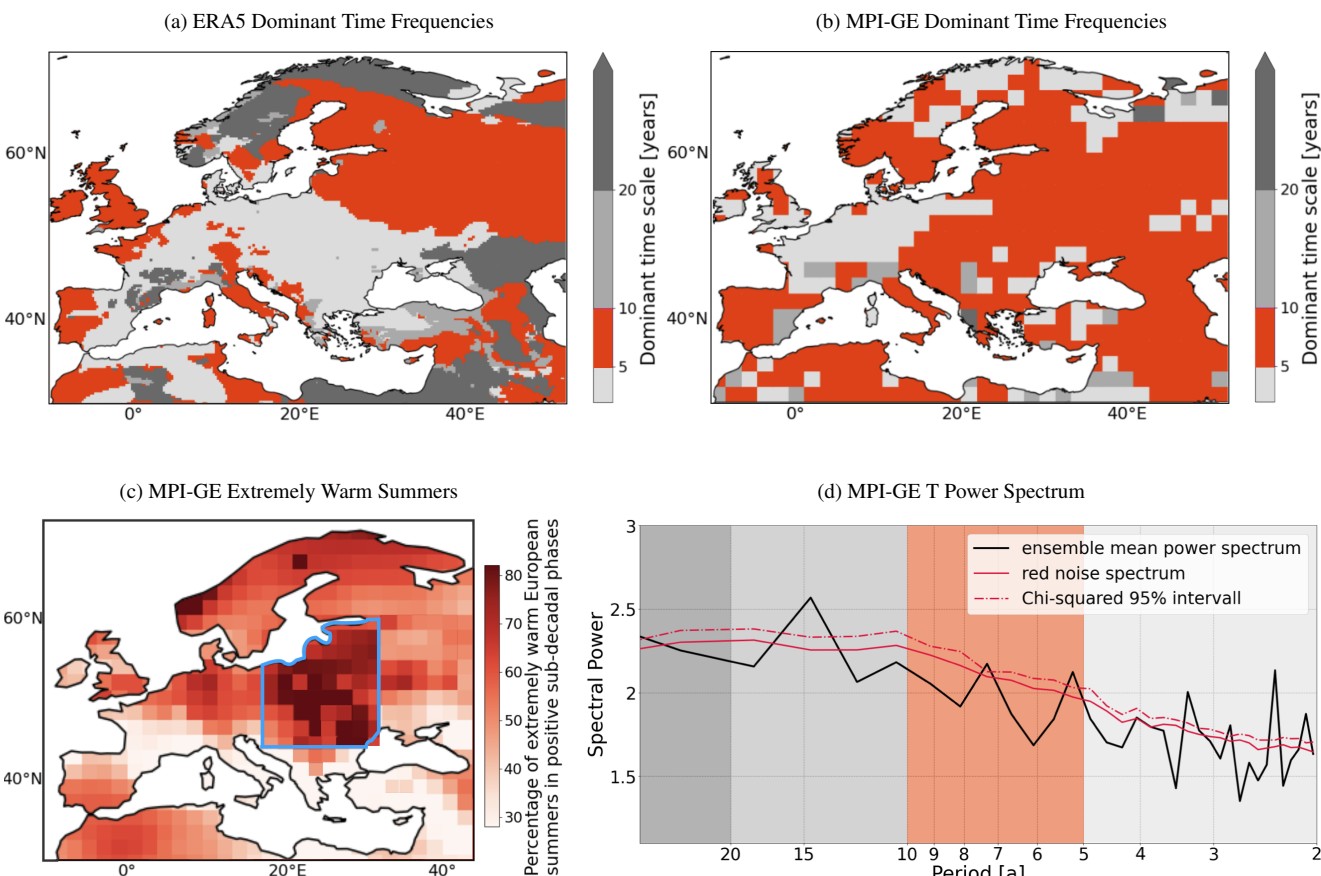

**Figure 1.** Dominant time frequencies and their relation to extremely warm European summersheat extremes. (a),(b) Cross-spectral analysis, performed using the multi-taper method, showing the dominant time scales of European surface air temperature variability in (a) ERA5 and (b) MPI-GE (see Methods). Color shading in years. (c) Percentage of all heat extremes (T>90th percentile) occurring in a positive bandpass filtered phase (T$bandpass$ >0) per grid-point in MPI-GE. The blue box defines the region of interest for further analysis (Central Europe, ~15°-35°E; 45°-60°N). (d) Power spectrum of Central European (spatial mean of blue box) surface air temperature (black line) in MPI-GE (averaged over all ensemble member spectra). The significance is shown via a red-noise spectrum (solid red line) and the chi-squared 95% interval (dashed red line). The background is color-coded according to the time intervals in (a,b). Period 1950-2022.

## 3.2 The North Atlantic Ocean and Extremely Warm European Summers

We start with our investigation with the North Atlantic ocean-atmosphere latent and sensible heat fluxes for lags up to three years prior to an extreme summer, to examine the North Atlantic Ocean long-term variability could drive the sub-decadal variability in extremely warm European summers (Fig. 2a).

At lag 0, when anomalies in the North Atlantic Ocean occur in the same year as the extreme summer, we find high anomalies, reaching up to 20% of the total variability, in the western part of the North Atlantic Ocean (30°-60°W; 25°-40°N), as well as in the north-eastern part of the North Atlantic Ocean (15°-25°W; 50°-70°N). These high positive anomalies in the North Atlantic Ocean, which indicate an above-average heat flux from the ocean to the atmosphere during extremely warm European summers and associated warming of the atmosphere, can be traced back several years prior to the extreme.

Although the global anomaly pattern suggests some relation to other long-term climate variability modes of the Pacific Ocean, such as the Pacific Decadal Oscillation and Tripolar Pacific Index (Fig. 2b), further analysis shows that e.g. ENSO does not drive the pattern described here (Table A1). The fraction of extremely warm European summers during the different ENSO phases is consistently low for different lags. Whereas, the fraction of extremely warm European summers strongly relies on the state of the North Atlantic oceanic variables. This means that no specific ENSO phase (El-Nino, La-Nina, Neutral) can be concretely associated with extremely warm European summers on sub-decadal time scales. Whether this relationship is coincidental and caused by an extraneous process (Cane et al., 2017), or whether this response is indicating a dynamical relationship between processes in the North Atlantic Ocean and the occurrence of extremely warm European summers, is further investigated in the following.

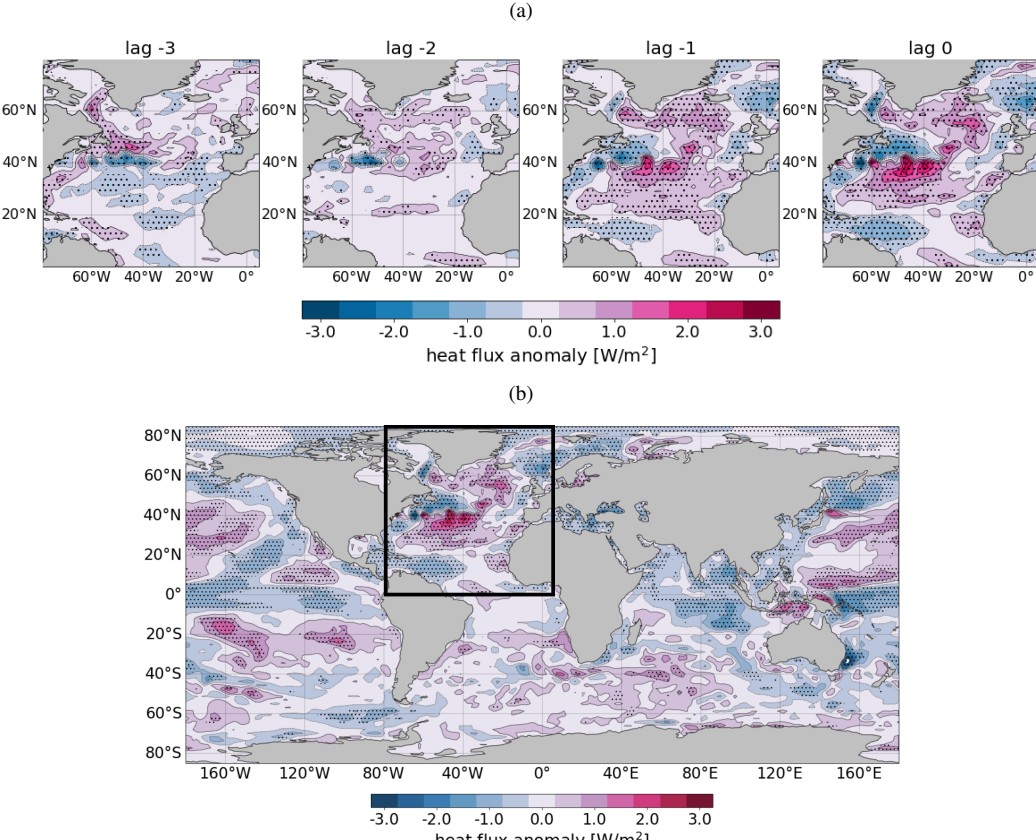

**Figure 2.** Anomaly of 5-10 year bandpass-filtered Atlantic heat flux (latent + sensible) variability in MPI-GE for (a) different lags prior to extremely warm European summers and (b) lag 0 as a global map. Positive values indicate heat flux into the atmosphere. Values in $\mathrm{Wm}^{-2}$, lags in years. Dots denote significance at a 95% confidence level. Period 1950-2022. For comparison the standard deviation of the year-to-year variation is of the order of $14\,\mathrm{Wm}^{-2}$, which means that the highest values in the figure correspond to around 20% of the total variability.

## 3.3 Influence of North Atlantic Ocean Heat Accumulation on Extremely Warm European Summers

We test if the oceanic variability in the North Atlantic Ocean can influence atmospheric circulation patterns via heat accumulation and release. Therefore we evaluate the relationship between North Atlantic Ocean inertia and extremely warm European summers. First, we analyze the ocean heat content, which influences the temperature difference between the ocean and atmosphere and thus alters the rate of heat exchange and is therefore a driver for the ocean-atmosphere heat flux. Here, we investigate anomalies of the 0-700m averaged, 5-10 year bandpass-filtered ocean heat content (Fig. 3a).

Starting around three years prior to extremely warm European summers, ocean heat content anomalies change from negative to positive all along the North Atlantic current, indicating an accumulation of heat in northern part of the subtropical gyre. For lag 0, these anomalies reach up to 25% of the total variability of the ocean heat content.

The ocean heat content is controlled by the meridional ocean heat transport, which describes the movement of heat energy
from one region of the ocean to another and can lead to changes in the ocean heat content in different regions over time. Here,
further insight into the dynamics of the North Atlantic Ocean subtropical and subpolar region is provided by the 5-10 year
bandpass-filtered ocean heat transport and its decomposition into a gyre- and meridional circulation part (Fig. 3b; calculated
independently, see Ghosh et al. (2023)). The anomalies of the 5-10 year bandpass-filtered ocean heat transport reveals positive
anomalies of the meridional heat transport around 20°N from two years prior to extremely warm summers onward. A substantial
proportion of these positive anomalies of the meridional heat transport is not compensated by ocean heat transport changes at
40°N. Here, due to the increased net heat transport around 40°N, the ocean heat content in that region will increase, leading
to the previously described accumulation of ocean heat content. The accumulated heat is released at lag 0, mainly through the
gyre ocean heat transport around 65°N. Altering the temperature gradient between the ocean and the atmosphere, this heat
release matches in turn with the positive ocean-atmosphere heat flux anomaly around 50-70°N (Fig. 2).

The ocean heat transport is influenced by the direction and strength of the horizontal oceanic currents, characterized by the
barotropic stream function. The barotropic stream function refers to the circulation of ocean currents at a certain depth, where
the flow is primarily influenced by pressure gradients. Changes in the barotropic stream function can indicate shifts in the paths
and intensity of ocean currents. As a result, the direction and strength of heat transport in the ocean may be affected. This, in
turn, leads to changes in the distribution of ocean heat content across different regions. Thus, the barotropic stream function
provides further knowledge about the the paths of ocean currents (Fig. 3c). Starting from three years prior to an extremely warm
European summer, negative anomalies of the barotropic stream function occur in the northern part of the subtropical gyre, indi-
cating a North Atlantic current weaker than its normal state, leading to a smaller horizontal volume transport and a southward
shifted subpolar gyre boundary around three years prior to an extremely warm summer. The anomalies of the barotropic stream
function change sign to positive values about one year prior to extremely warm summers, indicating strengthening of the North
Atlantic current and associated greater horizontal volume transport. Moreover, the North Atlantic current shifts by a few de-
grees north compared to the mean state, indicating a volume transport into higher latitudes via the North Atlantic current. This
increased northern horizontal volume transport together with the transition of the ocean heat content indicates the accumulation
of heat along the northern branch of the subtropical gyre.

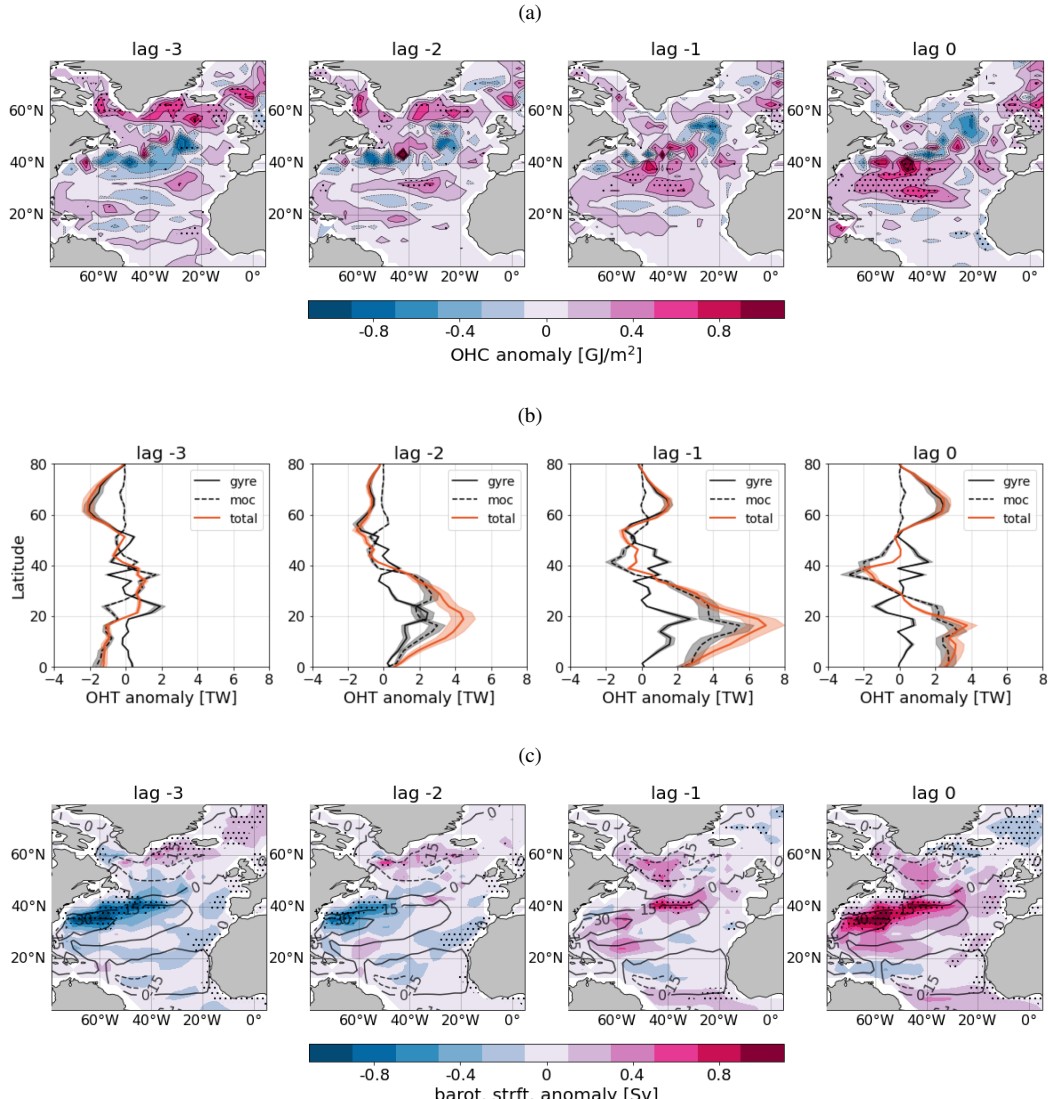

**Figure 3.** Extremely warm European summers and their relation to ocean quantities. (a) Upper 700m ocean heat content. Anomaly of 5-10 year bandpass-filtered ocean heat content variability in MPI-GE for different lags prior to extremely warm European summers, values given in GJm$^{-2}$. For comparison the standard deviation of the year-to-year variation is of the order of $4.3\,\mathrm{GJm}^{-2}$, which means that the highest values in the figure correspond to around 25% of the total variability. (b) Ocean heat transport. Anomaly of 5-10 year bandpass-filtered ocean heat transport variability in MPI-GE for different lags prior to extremely warm European summers, values given in TW. For comparison the standard deviation of the year-to-year variation is of the order of $50\,\mathrm{TW}$, which means that the highest values in the figure correspond to around 15% of the total variability. (c) Barotropic stream function. Anomaly of 5-10 year bandpass-filtered barotropic stream function variability in MPI-GE for different lags prior to extremely warm European summers, values given in Sv. Contour lines indicate the mean state of the barotropic stream function, values given in Sv. For comparison the standard deviation of the year-to-year variation is of the order of 8 Sv, which means that the highest values in the figure correspond to around 15% of the total variability. All lags are given in years. Dots (a, c) and shadings (b) denote significance at a 95% confidence level. Period 1950-2022.

## 3.4 Atmospheric Pathway leading to Extremely Warm European Summers

Three years prior to an extremely warm European summer, heat accumulates along the North Atlantic current. This heat is subsequently released into the atmosphere at lag 0. Here, we explain the atmospheric response bridging the ocean heat accumulation with the European summer climate.

The anomaly of the 5-10 year bandpass-filtered atmospheric temperature reveals positive temperature anomalies especially in higher latitudes around 50/60°N (Fig. 4a). These temperature anomalies spatially fit to the previously located anomalies of the ocean heat content and resemble the heat accumulation shown in the previous section. Based on this dynamical linkage we conclude that the ocean is warming the atmosphere via the ocean-atmosphere heat flux rather than the atmosphere is cooling the ocean. Our conclusion is also supported by the positive sign of the heat flux anomaly, indicative of heat flux transfer from the ocean to the atmosphere. The transfer of heat from the ocean to the atmosphere is strong enough that its signal reaches up to 200 hPa altitude, with a peak in the range of 400-600 hPa. This warming of the tropospheric high latitudes provides a decrease of the meridional temperature gradient and results in a reduction of wind shear due to the thermal wind balance. This leads to a weakened jet stream in the years with extremely warm summers compared to years without extremely warm European summers. In addition, the average position of the jet stream is shifted northward during extremely warm European summers, this northward shift indicates the advance of subtropical air masses into higher latitudes (Fig. 4; orange contour lines).

5-10 year bandpass-filtered sea level pressure anomalies, reveal a structure of a Scandinavian Blocking, which can be identified considering years with and without extremely warm summers (Fig. 4b). The Scandinavian Blocking can drive heat extremes over Central Europe (Spensberger et al., 2020), and connects the sub-decadal North Atlantic Ocean heat accumulation leading via specific atmospheric conditions to extremely warm summers over Central Europe. Additionally, some studies show that the weakening of wind speeds during extremely warm European summers can increase the probability of atmospheric blocking (Woollings et al., 2018), which would in turn increase the likelihood of heat extremes (Kautz et al., 2022). Here, we showed that the long-term accumulation of heat in the North Atlantic Ocean lead to an above average ocean-atmosphere heat flux, which in turn can influence the atmospheric circulation and could further affect the occurrence of long-lasting high-pressure systems, favoring blocking.

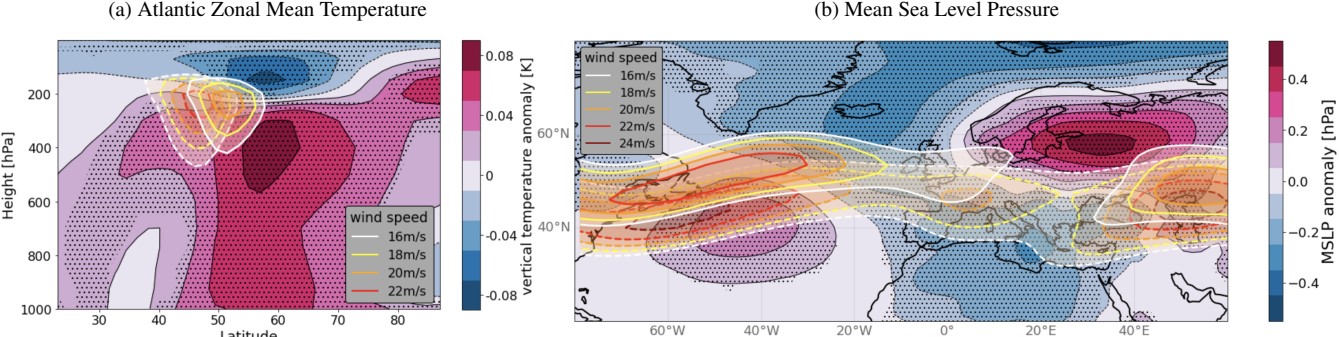

**Figure 4.** Extremely warm European summers and their atmospheric pathway. (a) Anomaly of 5-10 year bandpass-filtered Atlantic zonal mean temperature variability in MPI-GE during extremely warm European summers (lag0), values given in K. For comparison the standard deviation of the year-to-year variation is of the order of 0.3 K, which means that the highest values in the figure correspond to around 30% of the total variability. (b) Anomaly of 5-10 year bandpass-filtered mean sea level pressure variability in MPI-GE during extremely warm European summers (lag0), values given in hPa. For comparison the standard deviation of the year-to-year variation is of the order of 3 hPa, which means that the highest values in the figure correspond to around 15% of the total variability. The orange contour lines indicate the mean position of the jet stream (given by the mean zonal wind speed over 200-300 hPa) averaged over years showing an extremely warm European summer (solid line) and years showing no extremely warm summer (dashed line), values given in m/s. Dots denote significance at a 95% confidence level. Period 1950-2022.

## 4 Discussion and Conclusion

The North Atlantic Ocean heat accumulation impacts the occurrence of extremely warm summers over Central Europe on sub-decadal timescales. Using MPI-GE, we show that starting several years prior, anomalies of the ocean heat transport and associated ocean heat content changes result in ocean-atmosphere heat flux anomalies leading to extremely warm European summers.

These positive anomalies of the ocean heat transport, as well as ocean heat content, lead to an intensification of the North Atlantic current and accumulation of heat content along the subtropical gyre. This accumulated heat content is released mainly through the ocean heat transport by the subpolar gyre to the atmosphere during extremely warm European summers. The released heat in turn leads to a displacement of the jet stream and enhanced atmospheric blocking conditions.

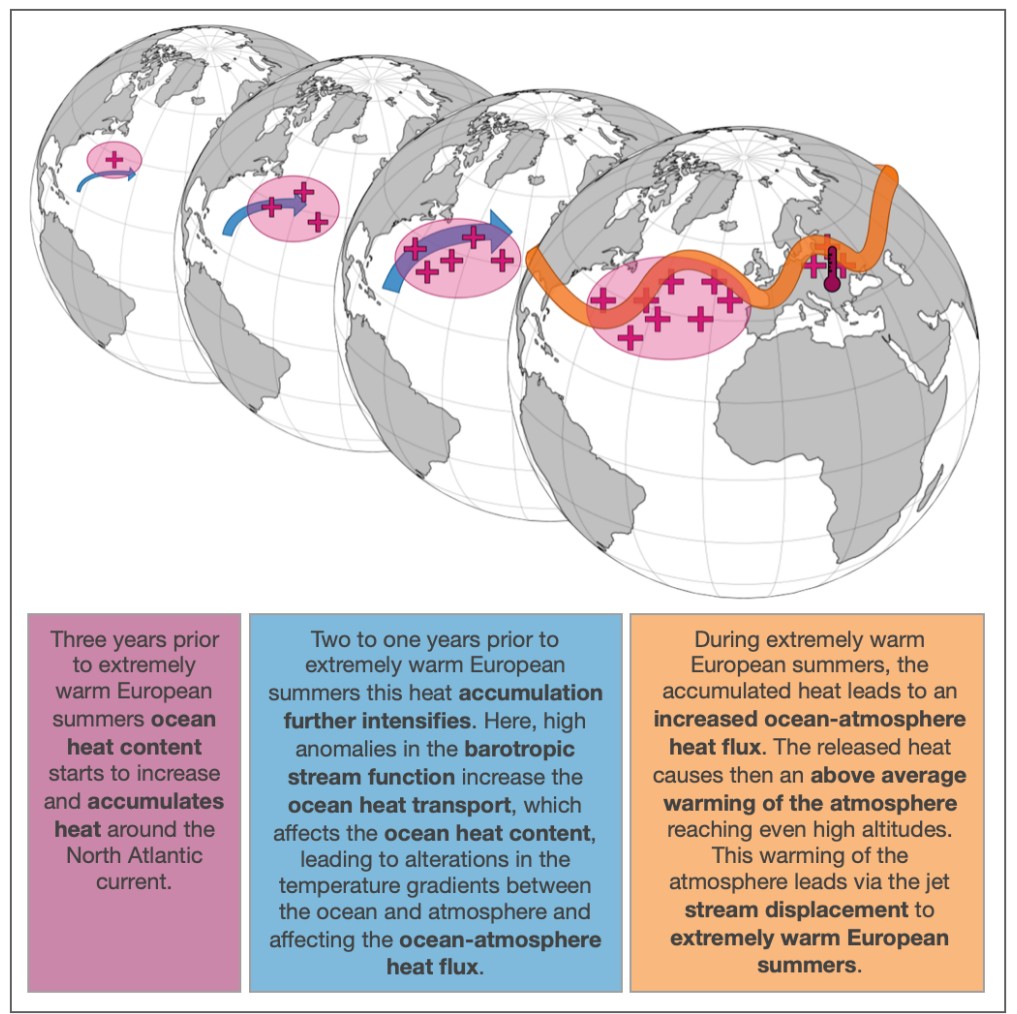

**Three years prior to extremely warm European summers ocean heat content** starts to increase and **accumulates heat** around the North Atlantic current.

**Two to one years prior to** extremely warm European summers this heat **accumulation further intensifies**. Here, high anomalies in the **barotropic stream function** increase the **ocean heat transport**, which affects the **ocean heat content**, leading to alterations in the temperature gradients between the ocean and atmosphere and affecting the **ocean-atmosphere heat flux**.

**During extremely warm** European summers, the accumulated heat leads to an **increased ocean-atmosphere heat flux**. The released heat causes then an **above average warming of the atmosphere** reaching even high altitudes. This warming of the atmosphere leads via the jet **stream displacement** to **extremely warm European summers**.

**Figure 5.** Schematic sketch illustrating the described mechanism. The blue arrows illustrate the increasing North Atlantic current; the pink crosses indicate the increase of ocean heat content and accumulation of heat; and the orange belt illustrates the jet stream. The thermometer at lag 0 illustrates the extremely warm European summers.

Although we focus on three years prior to the extremely warm summers, there is evidence potentially linking this mechanism to a fully coupled atmosphere-ocean cycle in the North Atlantic Ocean evolving in a 7-10 year period. Such oscillating behavior has been identified in a number of quantities involving observed sea surface temperatures and Gulf Steam indices (Czaja and Marshall, 2001; McCarthy et al., 2018), or heat content and overturning stream functions (Martin et al., 2019), or for the North Atlantic Oscillation (NAO; Costa and Verdiere, 2002)). In fact, observations reveal that the European summer mean climate is ultimately connected to such a coupled atmosphere-ocean cycle (Müller et al., 2020). Comparable to our results, Martin et al. (2019) identified a similar atmosphere-ocean cycle using also the MPI-ESM in the low-resolution setup. Extending our analysis up to eight years prior to extremely warm summers is in line with their results, indicating the close relationship of the

185 occurrence of extremely warm European summers with the sub-decadal North Atlantic atmosphere-ocean cycle (see Figures A1-A3).

We find that the coupled oscillation in the North Atlantic Ocean influences the occurrence of very hot summers in Europe on sub-decadal time scales. However, on this timescale other modes of oceanic variability, such as the North Atlantic Oscillation, El Niño-Southern Oscillation, or Pacific Decadal Oscillation may also be influential. Observed decadal teleconnections between
190 the Pacific Ocean and North Atlantic Ocean have been shown e.g. in Müller et al. (2008). Global maps of heat flux during extremely warm European summers reveal negative anomalies within the tropical Pacific and patterns matching with the Pacific decadal variability (PDV; 90°-170°W/20°N), indicating an influence of other external drivers. However, a direct effect of PDV or ENSO on the sub-decadal occurrence of extremely warm European summers has not been found here. Additionally, our findings might also be impacted by other mechanisms that interact with each other and possibly lead to the occurrence of
195 extreme events over Europe. For example the North Atlantic Oscillation (NAO) and the Atlantic multi-decadal variability (AMV) exert significant influence on the occurrence of extreme events over Europe (Scaife et al., 2008; Qasmi et al., 2021). The NAO plays a crucial role in shaping weather patterns, contributing to the development of heatwaves and droughts, while variations in the AMV can impact atmospheric circulation patterns, influencing the frequency and intensity of extreme events. Whether further climate modes have an impact on the proposed mechanism is beyond of the scope of this manuscript, and
200 should be further explored.

Here, we focus on extremely warm European summers associated with the decadal atmosphere-ocean coupling in the North Atlantic Ocean. However, given that the coupled cycle appears over several years, we expect that there is not only an influence on the summertime, but also for other seasons and respective extreme conditions, and further variables relevant for heat extremes, such as the daily maximum temperature. Furthermore, the presented process of ocean heat distribution changes at
205 multi-year lead times prior to an extreme event sets prospects to enhance the predictability of European climate and extremes. Multi-year prediction skill of European climate has been achieved on continental scale (Smith et al., 2020). However, an extension to predict extreme conditions has so far not fully been established (Borchert et al., 2019).

Moreover, the investigated sub-decadal extreme heat variability implies increased risk of heat-related socioeconomic and ecological impacts, in addition to year-to-year variability and rising temperatures due to increasing GHG concentrations. Due
to the prominence of the sub-decadal variability and due to the severity of the impacts, a deeper understanding of the sub-decadal processes leading to such extremely warm summers is crucial for reducing the uncertainties in both attribution and prediction of high-impact events, which in turn facilitates preparedness and the efficiency of adaptation and mitigation efforts.

Lastly, this is a single model study which allows us to delve deeper into specific processes and model intricacies, which can contribute to model improvement and process understanding. Replicating this analysis for different climate models would
be of great importance to sample potential model uncertainty in these results and help us gain further understanding of this mechanism.

Moreover, the investigated sub-decadal extreme heat variability implies increased risk of heat-related socioeconomic and ecological impacts, in addition to the year-to-year variability and rising temperatures due to increasing GHG concentrations. Due to the prominence of the sub-decadal variability and due to the severity of the impacts, a deeper understanding of the

220 sub-decadal processes leading to such extremely warm summers is crucial for reducing the uncertainties in both attribution and prediction of high-impact events, which in turn facilitates preparedness and the efficiency of adaptation and mitigation efforts.

*Data availability.* Further simulation and download details for MPI-GE data can be found on our website (https://www.mpimet.mpg.de/en/grand-ensemble/). ERA5 data are available from the European Centre for Medium-Range Weather Forecasts (ECMWF) (https://www.ecmwf.int/en/forecasts/dataset/ecmwf-reanalysis-v5).

# Appendix A

## A1 Barotropic Stream Function and Subpolar Gyre

The fact that no pronounced anomalies of the barotropic stream function can be found in the area of the subpolar gyre is probably related to the filter method we chose. The 5-10 year bandpass filter filters out all signals that occur on larger or smaller time scales. According to Nigam et al. (2018) the sub-polar gyre is subject to decadal variations of about 14 years and time scales which are not further relevant for our analysis.

## A2 Link to fully coupled atmosphere-ocean cycle

Analyzing longer lags, in this case lag -7 to 0, prior to extremely warm European summers shows that the described mechanism can be seen as attached to a fully coupled atmosphere-ocean cycle evolving in a 7-10 year period (Figure A1-A3). Such oscillating behavior, without linkage to European summer climate, has been identified and described in previous studies (Czaja and Marshall, 2001; McCarthy et al., 2018; Martin et al., 2019). In Martin et al. (2019) a NAO-like wind-driven forcing steering dipolar ocean overturning anomalies are associated with a contraction and weakening of the sub-polar gyre (cf their figure 6). In the following years, the barotropic stream function reveals a poleward shift and a strengthening of the North Atlantic current at the same time accumulates ocean heat content (cf. their figure 7). The barotropic stream function in MPI-GE prior to heat extremes similarly illustrates strengthening of the North Atlantic current and accumulation of heat. For longer lags a phase reversal is apparent, similar to the oscillatory behavior of the coupling as described in Martin et al. (2019).

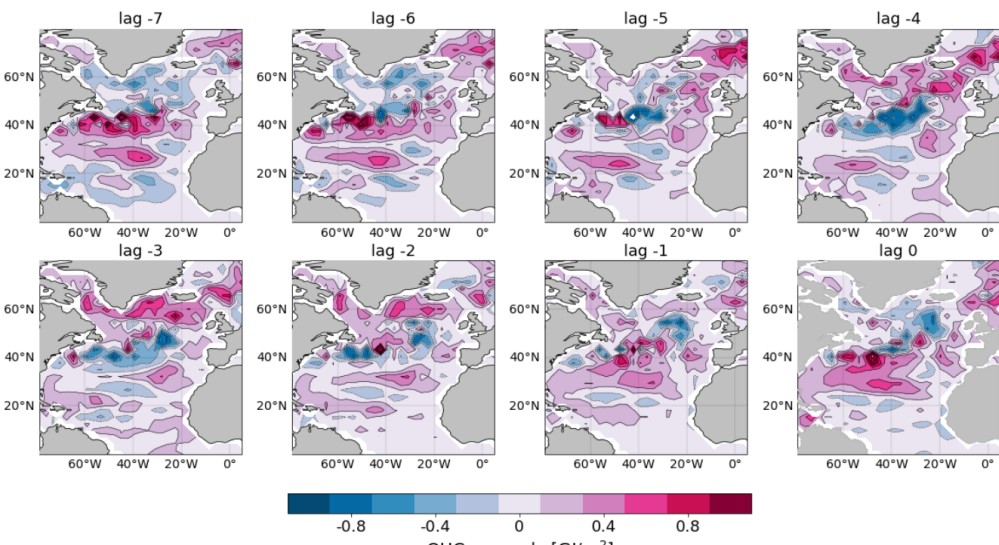

**Figure A1.** Shift of the ocean heat content signal. Anomaly of 5-10 year bandpass-filtered ocean heat content (0-700m/30-60°W) variability in MPI-GE for different lags prior to heat extremes. Period 1950-2022.

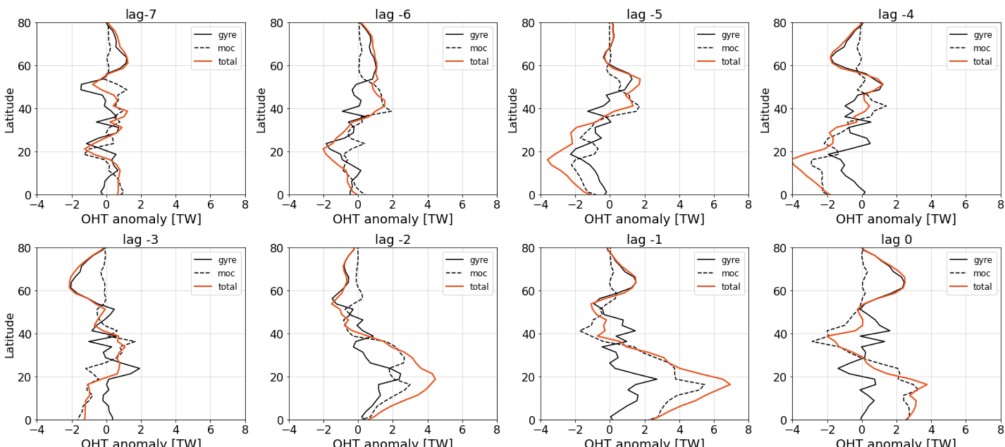

**Figure A2.** Shift of the ocean heat transport signal. Anomaly of 5-10 year bandpass-filtered ocean heat transport variability in MPI-GE for different lags prior to heat extremes. Period 1950-2022.

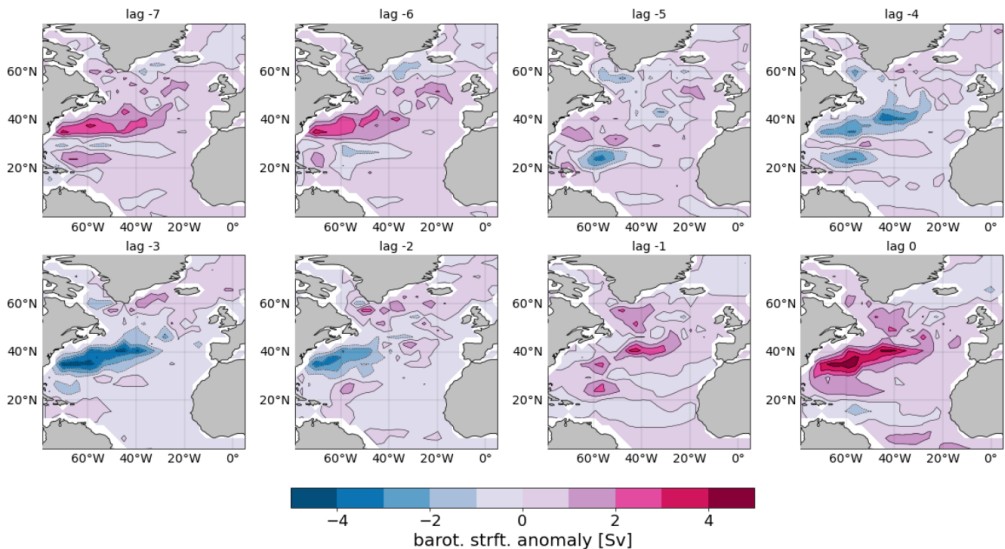

**Figure A3.** Shift of the barotropic stream function signal. Anomaly of 5-10 year bandpass-filtered barotropic stream function variability in MPI-GE for different lags prior to heat extremes. Period 1950-2022.

## A3 Influence of the El Nino Southern Oscillation

Many studies have examined the influence of the El-Nino Southern Oscillation (ENSO) on European temperatures and also heat extremes (Martija-Díez et al., 2021). However, ENSO does not seem to play a dominant role for the mechanism studied here. On the one hand, the fraction of extremely warm European summers during the different ENSO phases (El-Nino, La-Nina, Neutral) is consistently low for different lags, so that no specific ENSO phase can be concretely linked to extremely warm European summers on sub-decadal time scales (Table A1). For this analysis, we defined ENSO phases by SSTs exceeding a

threshold of $\pm$ one standard deviation in the Nino-3.4 region; however by testing other thresholds we verify that our conclusion is not threshold sensitive. Our statement that the extremely warm European summers are mainly driven by North Atlantic Ocean heat accumulation and not ENSO is further supported by the proportion of extremely warm European summers during positive/negative anomalies of the barotropic stream function in the North Atlantic Ocean. Here, a clear correlation between both can be seen for different lags. While in lag 0 the extremely warm summers are mainly associated with positive anomalies of the barotropic stream function, these are in lag -4 mainly associated with negative anomalies of the barotropic stream function. Furthermore, for our study, no specific ENSO phase seem to be linked to specify anomalies of the barotropic stream function. In addition, the heat flux anomalies during extremely warm European summers show no typical ENSO pattern in the Nino-3.4 region (Fig. 2b). However, positive anomalies matching the North Pacific Index (NPI) could be found around 90°-170°W/20°N, indicating an influence of external drivers that, although beyond of the scope of this study, should be further explored.

**Table A1.** Fraction of events that coincide with extremely warm European summers in MPI-GE. Period 1950-2022. The percentages are given by the ratio between the number of events (e.g. El-Nino events) during extremely warm European summers and the number of all occurring events.

| | Fraction of events that coincide with an extremely warm European summer [%] | | |
|---|---|---|---|
| | Lag -4 | Lag -2 | Lag 0 |
| **El-Nino events** | 9.1 | 8.3 | 7.5 |
| **neutral events** | 8.2 | 8.2 | 8.3 |
| **La-Nina events** | 7.2 | 7.7 | 8.0 |
| **positive barotropic stream function anomaly** | 4.5 | 7.1 | 13.2 |
| **negative barotropic stream function anomaly** | 11.7 | 9.2 | 3.0 |

*Author contributions.* L.W. did the analysis in correspondence with W.A.M., L.S.G., and D.M. and wrote the initial version of the manuscript. All authors commented on the manuscript. W.A.M., L.S.G., and D.M. provided guidance on the overall direction of the work.

*Competing interests.* The authors declare no competing financial interests.

*Acknowledgements.* This work was funded by the International Max Planck Research School on Earth System Modelling (IMPRS-ESM; L.W.), the ClimXtreme project DecHeat (Grant number 01LP1901F; L.S.G., W.A.M.), the European Union's Horizon Europe Framework Programme under the Marie Skłodowska-Curie grant agreement (101064940; L.S.G.), and the JPI Oceans NextG-Climate Science-ROADMAP project (01LP2002A; D.M.). We thank Luis Kornblueh, Jürgen Kröger, and Michael Botzet for producing the MPI-GE sim-
265 ulations we used for our analysis, Rohit Ghosh for providing the MPI-GE ocean heat transport data we used for our analysis, and Johann Jungclaus for reviewing the manuscript at an earlier stage. Additionally, we acknowledge the Swiss National Computing Centre (CSCS) and the German Climate Computing Center (DKRZ) for providing the necessary computational resources. Furthermore, we would like to thank Olivia Martius, as well as two anonymous reviewers for their valuable comments within the review process.

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
