# Peer review of "Extremely Warm European Summers preceded by Sub-Decadal North Atlantic Ocean Heat Accumulation"

_EGUsphere, 2023_

## Author Comment (AC1)

**Response to the editor:**

We highly appreciate and are very thankful for the time and effort that was invested in reviewing our manuscript. The detailed and constructive feedback will help us to improve the manuscript. In the following, we provide an answer to each comment brought up by the editor and reviewers. The original comments are in italic red while our responses are in black.

*Title: Recommend replacing with the word "driver" with "preceded," driver implies causality which is not explicitly shown in this paper.*

We agree with the editors suggestion and plan to change "Extremely Warm European Summers driven by Sub-Decadal North Atlantic Heat Inertia" to "Extremely Warm European Summers preceded by Sub-Decadal North Atlantic Heat Accumulation".

*L 27 please specify "different ocean-related quantities"*

The different ocean related quantities refer to the overturning stream function, ocean heat content, barotropic stream function, and sea surface temperature. In order to specify this term, we will change this sentence to "The variability in the North Atlantic region for different ocean-related quantities, such as the overturning stream function, ocean heat content, barotropic stream function, and sea surface temperature, indicates a fully coupled atmosphere-ocean cycle with a period of about 7-10 years".

*L59 please explain the taper method*

The multi-taper method after Arthun et al. (2018) is a spectral analysis technique used to estimate the dominant frequency of time series data by decomposing the data into a set of orthogonal tapers and computing a set of spectral estimates. The dominant frequency is then identified as the highest spectral peak in the resulting spectrum, which helps to characterize and understand the dominant oscillatory patterns and variability of the data over time.
For clarification, we will add this explanation to the corresponding paragraph.

*L68 Please also control for the false discovery rate -> see Wilks https://doi.org/10.1175/BAMS-D-15-00267.1*

We want to thank the editor for this suggestion. We will calculate the p-value from our 1000 bootstraps and control for the false discovery rate (equation 3 in Wilks et al. 2016) with a chosen control level of $\alpha_{FDR}= 0.1$. Preliminary tests show no substantial effect on our results, e.g. for the heat flux anomalies:

[Figure]

*L72ff Please add equations to the explanations for clarification*

We agree with the editor and will add physical equations for completion of the equation that already exists in written out form.

*L82 Please mention how the anomalies are defined?*

The anomalies are calculated by removing the long-term climatological average defined in the period 1850 to 2022.

*L105 mechanism --> pattern*

We will change "mechanism" to "pattern".

*L112 Please explicitly formulate the suggested causal relationship*

In this case, the causal relationship refers to the relationship between North Atlantic heat inertia (cause) and extremely warm European summers (effect). In particular, we analyze if the oceanic variability in the North Atlantic can influence atmospheric circulation patterns via heat accumulation and release, which in turn could lead to extremely warm European summers. We will rewrite this sentence to be more precise, such as "To test the relationship between ocean inertia and extremely warm European summers, we analyze if the oceanic variability in the North Atlantic can influence atmospheric circulation patterns via heat accumulation and release which in turn could lead to extremely warm European summers. First, we analyze the ocean heat content, which influences the temperature gradient between the ocean and atmosphere and thus alters the rate of heat exchange and is therefore a driver for the ocean-atmosphere heat flux."

*L113 gradient --> difference*

We will change "gradient" to "difference".

*L132 Please explain for a non-oceanographer how the barotropic stream function can alter the path of the currents? The stream function is per se only a diagnostic measure.*

We agree with the editor that the barotropic stream function is per se only a diagnostic measure and therefore, it can not alter the path of the currents or accelerate/slowdown them. We will revise the manuscript accordingly.

*L149 Why do we know that the heat is coming from the ocean?*

We thank the reviewer for bringing this up. Since the mechanism we analyzed evolves over several years in the North Atlantic ocean, which includes the accumulation of heat, we conclude that the ocean is warming the atmosphere via the ocean-atmosphere heat flux rather than the atmosphere is cooling the ocean. Our conclusion is also supported by the positive sign of the heat flux anomaly (indicative of heat flux transfer from the ocean to the atmosphere).

*L156 How do you know that this is a block and not just a high-pressure system?*

The fact that we analyze summertime means, together with the weaker jet stream, leading to more stationary weather conditions, lead to the conclusion that this is at least a long-living high pressure system, favoring a blocking situation. A detailed analysis is in fact pending.

*L158 How does it confirm the connection between the sub-decadal variability – please explain in more detail.*

We agree with the editor, that this sentence needs to be expanded to explain the mentioned connection, which refers to the identification of the link from North Atlantic ocean heat inertia over specific atmospheric conditions (such as Scandinavian Blocking) to extremely warm European summers. We will extent and rewrite the sentence accordingly, such as "The Scandinavian Blocking can drive heat extremes over Central Europe (Spensberger et al., 2020)), and confirms the connection between sub-decadal North Atlantic ocean heat inertia leading to specific atmospheric conditions to extremely warm summers over Central Europe."

*L160 The link between weaker jet streams and blocks is still contested, there are also arguments that a weaker temperature gradient results in a reduced blocking frequency see e.g., https://agupubs.onlinelibrary.wiley.com/doi/full/10.1002/2014GL060764*

We thank the editor for bringing this to our attention, to reflect this, we will soften our statement to "Additionally, some studies show that the weakening of wind speeds during extremely warm European summers can increase the probability of atmospheric blocking (Woollings et al., 2018), which would in turn increase the likelihood of heat extremes (Kautz et al., 2022)."

*L161 I disagree with the statement that you show how the heat fluxes lead to blocking, this point needs to be further substantiated.*

We have to apologize for this confusion. Our statement is perhaps a bit too overstated and we plan to tone down our wording. Here, we want to say that the heat flux anomalies can influence via the warming the atmosphere the atmospheric circulation and could thus affect the occurrence of long-lasting high-pressure systems such as blocking. Change to "Thus, we show how long-term North Atlantic heat inertia leads to accumulation of heat and an above average ocean-atmosphere heat flux, which can influence via the warming the atmosphere the atmospheric circulation and could thus affect the occurrence of long-lasting high-pressure systems such as blocking".

*Figure 1 Please indicate how many events contribute to the composites*

We have to apologize for the confusion. In figure 1 we actually do not show composites. Figure 1a and figure 1b show the dominant time scales of European surface air temperature variability in MPI-GE and ERA5 by using a cross-spectral analysis, figure 1c shows the amount of extremely warm European summers on sub-decadal time scales, and figure 1d the power spectrum of Central European surface air temperature. For our analysis we can investigate 730 extremely warm European summers in MPI-GE, we will add this number to the corresponding text.

We must apologize for this confusion. This refers to the number of extreme summers that occur during a simultaneous positive sub-decadal phase in European temperatures per grid cell. The number of extremely warm European summers (T>90th percentile) per ensemble member refers to the entire time period (73 years). This means that from the approximately 1 in 10th summers that are extreme, as exceeding the 90th temperature percentile, as many as shown in the figure occur during a positive phase of the different regions. We can see that on average over all 100 members more extreme summers linked to sub-decadal (5-10 year) variations occur over Central Europe than in other regions. We agree with the editor that the unit of this figure is not completely clear,

therefore, we plan to revise the figure, for instance by indicating the percentage of extremely warm European summers occurring in positive sub-decadal phases per grid point, such as Figure A.

[Figure]

**Figure A:** Suggested new figure 1c, percentage of all heat extremes (T>90th percentile) occurring in a positive bandpass filtered phase (T_bandpass>0) per grid-point in MPI-GE. The blue box defines the region of interest for further analysis (Central Europe, ~15°-35°E; 45°-60°N).

*Figure 2: dots are very hard to see*

We agree that the dots are hard to see in a print-out. We will increase the dot size in order to increase their visibility.

---

## Author Comment (AC2)

**Response to the reviewer #1:**

We highly appreciate and are very thankful for the time and effort that was invested in reviewing our manuscript. The detailed and constructive feedback will help us to improve the manuscript. In the following, we provide an answer to each comment brought up by the editor and reviewers. The original comments are in italic red while our responses are in black.

*In this work, the mechanism of sub-decadal climate variability leading to extreme summer temperature events over Europe are investigated by using a Large Ensemble (LE) performed with the MPI-ESM model. The results are relevant in the context of climate prediction, the analyses are robust and well presented, as the quality of the figures is good. However, the language used in this manuscript is not precise and rigorous enough for a scientific article. There are many inaccuracies that make the text difficult to understand. Surprisingly the second part of the article, from section 3.3 is much better written than the first part.*
*Please, revise your manuscript and improve the precision of the language to make the text clearer. Below some examples and other remarks:*

We thank the reviewer for the comments and for sharing our view on relevance of this paper in the context of climate prediction. We would like to apologize for the confusion that has been caused by non-precise wording. In the following, we will go into more detail on how we intend to eliminate these and other sources of ambiguity.

*1.Abstract. The abstract is a very important part in the article. It needs to be well written. Please consider to rewrite the abstract to be more precise. Below an example of a rewritten abstract. This precision should be used throughout the text.*

*The internal variability of European summer temperatures has been linked to various mechanisms from sub-seasonal to multi-decadal timescales. However, the mechanisms controlling sub-decadal (< 10 years) variations remain unexplored. We find that sub-decadal time scales (3-5 years) dominate summer temperature variability over large parts of the European continent. We show that extremely warm summers over Europe, occurring in sub-decadal periods, are related by a strengthening of the Atlantic Ocean subtropical gyre, an increase of meridional heat transport, and an accumulation of ocean heat content over the North Atlantic several years prior to the extreme event episode. The ocean warming affects the ocean-atmosphere heat fluxes, leading to a weakening and northward displacement of the jet stream and increased probability of occurrence of atmospheric blockings over Scandinavia. Thus, our findings link the occurrence of extremely warm European summers to the thermal inertia of the North Atlantic Ocean, whose potential to improve the predictability of extremely warm summers several years ahead is of great societal interest, especially in a warming climate.*

We would like to express our appreciation for this very detailed suggestion on how exactly to rewrite the abstract. We will revise the abstract and the whole text regarding the clarity of our wording again.

1. *Line 16: "increase in variability" needs to be more precise.*

By "increase in variability" we refer to the increase in internal temperature variability, which, together with the general temperature increase, leads to more extreme European summers, as demonstrated by the cited studies. To clarify this, we plan to use "increase in internal temperature variability" instead of just "increase in variability".

2. *Line 21: Remove "as part of the large-scale…."*

We will remove "as part of the large-scale….".

3. *Line 25: "In particular" means what here ???*

"In particular" refers to "long memory mechanisms" at the beginning of the sentence. However, we agree, that this wording could be clearer, therefore we plan to split the sentence into two parts and mention „long-memory mechanisms" again in the second sentence. We plan to change " …variabilty, in detail such as…"  to "…variability. This means in detail long term mechanisms such as…".

4. *Different ocean-related quantities: Define*

The different ocean-related quantities refer e.g. to the overturning stream function, ocean heat content, barotropic stream function, and sea surface temperature. In order to specify this term, we plan to change this sentence to "The variability in the North Atlantic region for different ocean-related quantities, such as the overturning stream function, ocean heat content, barotropic stream function, and sea surface temperature, indicate a fully coupled atmosphere-ocean cycle with a period of about 7-10 years".

5. *Line 28: Please, include all the references at the end of the sentence and not in the middle. This is applicable to the rest of the article.*

We agree with the reviewer that references at the end of sentences enhance the readability and thus contribute to the clarity of the text. We will change this throughout the revised manuscript.

6. *Lines 31-33: Please rewrite, not very understandable.*

We will rewrite this section such as "This research aims to address these inquiries and present a comprehensive explanation for the occurrence of unusually hot European summers attributing it to the heat accumulation that occurs several years in advance. Our investigation concentrates on the exceptionally warm European summers that occur during multi-year periods."

7. *In the introduction there should be a reference to DCPP-C (AMV pace makers experiments) and their findings. I think that the findings of this project are relevant and need to be mentioned in this article.*

Thanks for this suggestion. We agree with the reviewer and plan to add a reference to the Decadal Climate Prediction Project (DCPP) and in particular their AMV pace maker experiments when mentioning the Atlantic multi-decadal variability, such as Boer et al. (2016), Ruprich-Robert et al. (2021), or Qasmi et al. (2021).

Boer, G. J., Smith, D. M., Cassou, C., Doblas-Reyes, F., Danabasoglu, G., Kirtman, B., Kushnir, Y., Kimoto, M., Meehl, G. A., Msadek, R., Mueller, W. A., Taylor, K. E., Zwiers, F., Rixen, M., Ruprich-Robert, Y., and Eade, R.: The Decadal Climate Prediction Project (DCPP) contribution to CMIP6, Geosci. Model Dev., 9, 3751–3777, https://doi.org/10.5194/gmd-9-3751-2016, 2016.

Qasmi, S., Sanchez-Gomez, E., Ruprich-Robert, Y., Bo., J., and Cassou, C.: Modulation of the Occurrence of Heatwaves over the Euro-Mediterranean Region by the Intensity of the Atlantic Multidecadal Variability, Journal of Climate, 34, 1099–1114, https://doi.org/10.1175/jcli-d-19-0982.1, 2021.

Ruprich-Robert, Y., Moreno-Chamarro, E., Levine, X. *et al.* Impacts of Atlantic multidecadal variability on the tropical Pacific: a multi-model study. *npj Clim Atmos Sci* **4**, 33 (2021). https://doi.org/10.1038/s41612-021-00188-5

8. *The term "Multi-year" is not precise. It is better to use sub-decadal.*

We agree and will change "multi-year" to "sub-decadal" in the revised manuscript.

9. *We use the largest ensemble…are you sure ?? I think that other LE also provide 100 members.*

Sorry for this confusion. To our knowledge there is one additional 100-member large ensemble available with historical and future scenario simulations from a fully coupled climate model, the CESM2-LE. Although this ensemble consists oft wo sets of 50-members under slightly different forcing conditions and provides only one future scenario, we will clarify this point in the manuscript to "one of the largest" to avoid ambiguities.

10. *What's a single model initial condition large ensemble?*

Single-model initial condition large ensemble refers to a term widely used in literature to describe a set of simulations from a single climate model under the same forcing conditions but starting from different initial conditions. We will clarify this point in the manuscript.

11. *North Atlantic ocean heat inertia, ocean is missing several times in the text after North Atlantic*

We thank the reviewer for this comment and will add "ocean" throughout the text.

12. *Remove "doing so", it's too familiar*

We will remove "doing so" throughout the text.

13. *Line 41: We confirm that the MPI-GE can represent sub-decadal temperature variability well…, where?*

This result can be found in section 3.1, where we analyze whether MPI-GE exhibits good representation of sub-decadal time scales in terms of their dominant time frequency with a cross-spectral analysis (fig. 1a compared to fig. 1b). and check how many extremely warm European summers can generally be found per grid point (fig. 1c).

14. *Remove" by identifying periods that increase….summers"*

We will remove this part of the sentence.

15. *….are done -> are performed*

We will change "are done" to "are performed".

16. *12-150 km à ??? what does it mean ?*

We will revise and expand the section on the ocean model grid/resolution. The MPI-M ocean model employed in our study (MPI-OM in GR15 resolution, Marsland et al. 2003) is formulated on a C grid and orthogonal curvilinear coordinates. To circumvent grid singularities at the geographical North Pole, the northern grid pole is shifted to Greenland, leading to high resolution in the Arctic and the high-latitude sinking regions. The horizontal resolution is about 1.5° on average and varies from a minimum of 12km close to Greenland to a maximum of 180km in the tropical Pacific. The model has 40

vertical non-equidistant z levels, of which 20 are distributed in the upper 700m. The coupled model does not employ flux adjustment.

> *Marsland, S. J., Haak, H., Jungclaus, J. H., Latif, M., & Röske, F. (2003). The Max Planck Institute global ocean/sea ice model with orthogonal curvilinear coordinates. Ocean Modelling, **5**, 91– 127.*

17. *ERA5 reanalysis -> include reference*

We already included a reference at the end of the sentence (see Hersbach et al., 2018 line 57).

18. *MPI-GE can represent the sub-decadal time scales well -> with respect to ???*

We analyze whether MPI-GE exhibits good representation of sub-decadal time scales in terms of their dominant time frequency compared to ERA5. This means whether the model accurately captures the frequency at which significant variations and patterns occur within periods of 5-10 years. We will clarify this sentence in the revised manuscript by adding "dominant time frequency".

19. *Please rewrite sentence Doing so, ….*

For more clarity and in order to avoid "doing so, …", we plan to combine this sentence with the previous sentence. "We perform the multi-tapering for all 100 ensemble members and each grid point to ascertain the dominant timescale, where the dominant timescale is given by the highest significant peak [Arthun et al., 2018, Ghil et al., 2002]."

20. *Remove "around the red noise"*

We will remove "around the red noise".

21. *Are you using daily data ? how the summer is defined here ?*

We are using monthly data averaged to seasonal summer means over June, July, and August. We plan to add the temporal resolution, as well as the study domain to the method section.

22. *The way the temperature extremes are defined in MPI-GE should be explained in the "Methods" section.*

We agree and plan to add the following definition "We define extremely warm European summers on sub-decadal timescales, hereafter referred to as extremely warm summers, as those summers which JJA mean temperature anomalies in the region between ~15°-35°E and 45°-60°N exceed their 90th percentile and additionally occur in a positive bandpass-filtered phase (pooled in time and ensemble (T>90th perc. and $T_{bandpass}>0$))" to the method section.

23. *I understand that the computations are done separately for each member in MPI-GE ? So what it is presented in the figures ? the ensemble mean of the quantity ?*

As the reviewer already mentioned we perform the spectral analysis individually for each ensemble member in MPI-GE. Afterwards we take the mean over all spectra for each grid point. For every grid point at least one peak was significant after a chi-squared 95% interval, if more than one peak for a specific grid point were significant, the more highly significant one was chosen. We hope that this answers the question and we plan to add the fact that we take the mean over all spectra afterwards to the multi-taper method section.

24. *Line 81. This is the conclusion of the paper, this is indeed what you want to prove…*

We agree that this statement might be misleading in this paragraph. We plan to rewrite this section and to move this sentence to the conclusions.

25. *Which variable is used as "temperature" to determine the extremes ? surface temperature, 2m temperature ?*

For all of our analyses we used the temperature at 2m height (T2m). We will add this to the revised manuscript.

26. *Line 87. In summary, sub-decadal timescales of 5-10….i do not understand the link to MEAN summer temperature since the focus are here the extremes…*

Here, we used "mean" since the cross spectral analysis is performed for the whole time series, not only extremes (a homogenous time series is needed for this kind of analysis). In the first figure only sub-figure 1c refers to extremely warm European summers, which are defined as those with mean summer temperatures above the 90th percentile. We have to apologize for this confusion and will add more text to explain the connection between the dominant time scales (fig. 1b) and number of extremely warm summers (fig. 1c).

27. *Lines 90-91, Figure 1.d. I see that there are significant peaks at periods of 15 and 2-4 years. I would not conclude that the sub-decadal is the dominant frequency here…*

We agree with the reviewer that the peak at about 15 years is the most dominant one, however, with this figure we want to show that the sub-decadal time scales also have significant peaks over Central Europe and thus we have a good reason to analyze them further. We plan to make this clearer in the text.

28. *Line 94. North Atlantic ocean heat content variability*

We thank the reviewer for this comment and will add "ocean" and "content", not only here, but throughout the text.

29. *Figure1.a and 1.b are not performed at the same spatial resolution, however it is mentioned that ERA5 is regridded on MPI grid, can you clarify ?*

We have to apologize for this confusion. We decided to maintain the figure as it is without regridded data and will remove this erroneous statement from the manuscript.

30. *Line98. Please be more accurate and define "years" throughout the text.*

We agree, that we haven't used "years" in an appropriate way in this manuscript. We plan to use "…up to three years prior to an extreme summer" instead of "…years prior to an extreme summer". We will change is throughout the manuscript.

31. *These high "positive" anomalies…*

We will add "positive" to this sentence.

32. *Line 104. Some relation to other long-term climate variability models over the Pacific… can you specify??*

In order to be more concrete, we will add the Pacific Decadal Oscillation and Tripolar Pacific Index as examples for possible other modes of ocean variability to this sentence, such as "Although the global anomaly pattern suggests some relation to other long-term

climate variability modes over the Pacific, such as the Pacific Decadal Oscillation and Tripolar Pacific Index …“.

33. *Please, rewrite lines 106-110.*

We will change these sentences to "Specifically, this means that no specific ENSO phase (El-Nino, La-Nina, Neutral) can be concretely associated with extremely warm European summers on sub-decadal time scales, meaning that the fraction of extremely warm European summers during the different ENSO phases is consistently low for different lags. Whereas, the fraction of extremely warm European summers strongly relies on the state of the North Atlantic oceanic variables. Whether this relationship is coincidental and caused by an extraneous process (Cane et al., 2017), or whether this response is  indicating a causal relationship, is further investigated in the following." Further, we will also study the influence of the Tripolar Pacific Index in addition to the influence of ENSO (see comment 34), we will add the results to the sentence above as we proceed.

34. *In addition to ENSO, have you consider the TPI (Tripolar Pacific Index) or IPV ? Anomalies shown in fig.2 look very much to TPI spatial structure.*

Thanks for this comment, so far we only tested a possible teleconnection with ENSO as one of the most well-known and influential teleconnections for European climate. However, of course it is worth mentioning that the ENSO teleconnection is just one of many factors that can influence European weather patterns, and other climate phenomena, such as the TPI, might also play a role. Therefore we agree with the reviewer and plan to test also the possible influence of TPI on our analysis.

35. *Line 120. Please, define the meridional ocean heat transport better.*

We will include in the revised manuscript a proper definition (including formulas) of the meridional ocean heat transport computation, both of the total and of the amoc/ gyre heat transport components.

36. *Line 145. "During the extreme." Please be more accurate.*

"During the extreme" refers to the year when the extremely warm European summer occurs, which is lag 0. We will change the old wording to this more precise wording.

37. *149. "temperature anomalies fit…" do you mean spatially fit ??*

Indeed we mean "spatially fit" here. We will add this to the corresponding sentence.

38. *Line 150. The heat transfer from the ocean to the atmosphere.*

We will rewrite this sentence as suggested by the reviewer: "The transfer of heat from the ocean to the atmosphere is so strong that its signal reaches up to 200 hPa altitude …".

39. *Line 152. Extremely European warm summers.*

As suggested by the reviewer, we plan to use "extremely warm European summers" instead of "heat extremes".

40. *Line 165: The North Atlantic OCEAN heat inertia…*

Thanks again for this comment. As already written, we plan to change "North Atlantic" to " North Atlantic ocean" throughout the text.

41. *Line 168. "together with an above average…" of what??*

"Together with an above average" refers to the volume transport of the barotropic stream function mentioned later in the sentence. We can see where this confusion comes from and will rewrite the sentence to make it clearer to what this refers to: "Positive anomalies of the ocean heat content, together with an above average horizontal volume transport as well as more northern horizontal volume transport of the barotropic stream function lead to a stronger North Atlantic current and accumulation of heat content."

42. *Lines 170-171. Please, specify where the "released heat" goes…to the atmosphere??*

Indeed the reviewer is right and the released heat goes into the atmosphere. We will add "to the atmosphere" in the corresponding sentence.

43. *Line 181. Rewrite: We find that the coupled oscillation in the North Atlantic prescribes extremely warm European summers on sub-decadal timescales.*

We will rewrite this sentence using more precise wording and tone it also down, such as "We find that the coupled oscillation in the North Atlantic influences the occurrence of very hot summers in Europe on sub-decadal time scales."

44. *Line 182. Other modes of ocean variability? Could you specify?*

In order to be more concrete, we will add the North Atlantic Oscillation, El Niño-Southern Oscillation, and Pacific Decadal Oscillation as examples for possible other modes of ocean variability to this sentence. "However, on this timescale other modes of oceanic variability, such as the North Atlantic Oscillation, El Niño-Southern Oscillation, or Pacific Decadal Oscillation may also be influential."

45. *There are other mechanisms leading to the occurrence of extreme events over Europe-Mediterranean, this should be mentioned in the text. Please see Qasmi et al. 2021 and references therein.*

We agree that extreme events are often the result of a combination of multiple factors and that also other mechanisms could lead to the occurrence of extreme events over Europe. We will add a paragraph discussing the influence of the NAO and AMV, such as "Additionally, our findings might also be impacted by other mechanisms that interact with each other and possibly lead to the occurrence of extreme events over Europe, e.g. the North Atlantic Oscillation (NAO) and the Atlantic multi-decadal variability (AMV) exert significant influence on the occurrence of extreme events over Europe (Scaife et al., 2008, Qasmi et al., 2021). The NAO plays a crucial role in shaping weather patterns, contributing to the development of heatwaves and droughts, while variations in the AMV can impact atmospheric circulation patterns, influencing the frequency and intensity of extreme events."

46. *The fact that the study was conducted with only one climate model should be discussed in the last section, as the mechanism described in this study could change from one model to another… so there is still uncertainty linked to the model used. We*

*are not sure that a different Large Ensemble performed with a different model would give the same results.*

We agree with the reviewer and plan to add a paragraph along the lines "This study is a single model study which allows us to delve deeper into specific processes and model intricacies, which can contribute to model improvement and process understanding. Replicating this analysis for different climate models would be of great importance to sample potential model uncertainty in these results and help us gain further understanding of this mechanism."

---

## Author Response (AR1)

**Response to the editor:**

We highly appreciate and are very thankful for the time and effort that was invested in reviewing our manuscript. The detailed and constructive feedback helped us to improve the manuscript. In the following, we provide an answer to each comment brought up by the editor and reviewers. The original comments are in italic red while our responses are in black.

*Title: Recommend replacing with the word "driver" with "preceded," driver implies causality which is not explicitly shown in this paper.*

We agree with the editors suggestion and changed "Extremely Warm European Summers driven by Sub-Decadal North Atlantic Heat Inertia" to "Extremely Warm European Summers preceded by Sub-Decadal North Atlantic Ocean Heat Accumulation".

*L 27 please specify "different ocean-related quantities"*

The different ocean related quantities refer to the overturning stream function, ocean heat content, barotropic stream function, and sea surface temperature. In order to specify this term, we changed this sentence to "The variability in the North Atlantic region has been shown to include a fully coupled atmosphere-ocean cycle with a period of about 7-10 years shown for different ocean-related quantities, such as ocean heat content and barotropic stream function".

*L59 please explain the taper method*

The multi-taper method after Årthun et al. (2018) is a spectral analysis technique used to estimate the dominant frequency of time series data by decomposing the data into a set of orthogonal tapers and computing a set of spectral estimates. The dominant frequency is then identified as the highest spectral peak in the resulting spectrum, which helps to characterize and understand the dominant oscillatory patterns and variability of the data over time.

For clarification, we added "We use a cross-spectral analysis, based on a multi-taper method to analyze if and where the MPI-GE and ERA5 can represent the sub-decadal time scales (Årthun et al., 2018). This multi-taper method is a spectral analysis technique to estimate the dominant time-frequency content of time series by decomposing the data into a set of orthogonal tapers and computing a set of spectral estimates. The dominant time-frequency is then identified as the highest spectral peak or mode in the resulting spectrum, which characterize the dominant oscillatory patterns and variability of the data over time. We perform …".

*L68 Please also control for the false discovery rate -> see Wilks https://doi.org/10.1175/BAMS-D-15-00267.1*

We want to thank the editor for this suggestion. We calculated the p-value from our 1000 bootstraps and control for the false discovery rate (equation 3 in Wilks et al. 2016) with a chosen control level of $\alpha_{FDR}= 0.1$. In comparison both techniques show no substantial effect on our results, e.g. for the heat flux anomalies:

[Figure]

*L72ff Please add equations to the explanations for clarification*

We agree with the editor and added physical equations instead of the equation in written out form.

*L82 Please mention how the anomalies are defined?*

The anomalies are calculated by removing the long-term climatological average defined in the period 1850 to 2022. We added "…calculated by removing the long-term climatological average defined in the period 1850 to 2022, …".

*L105 mechanism --> pattern*

We changed "mechanism" to "pattern".

*L112 Please explicitly formulate the suggested causal relationship*

In this case, the causal relationship refers to the relationship between the accumulation of heat in the North Atlantic Ocean (cause) and extremely warm European summers (effect). In particular, we analyze if the oceanic variability in the North Atlantic can influence atmospheric circulation patterns via heat accumulation and release, which in turn could lead to extremely warm European summers. We have rewritten this sentence to be more precise: "We test if the oceanic variability in the North Atlantic Ocean can influence atmospheric circulation patterns via heat accumulation and release. Therefore we evaluate the relationship between the accumulation of heat in the North Atlantic Ocean and extremely warm European summers. First, we analyze the ocean heat content, which influences the temperature difference between the ocean and atmosphere and thus alters the rate of heat exchange and is therefore a driver for the ocean-atmosphere heat flux".

*L113 gradient --> difference*

We changed "gradient" to "difference".

*L132 Please explain for a non-oceanographer how the barotropic stream function can alter the path of the currents? The stream function is per se only a diagnostic measure.*

We agree with the editor that the barotropic stream function is per se only a diagnostic measure and therefore, it can not alter the path of the currents or accelerate/slowdown them. We changed the paragraph to "The ocean heat transport is influenced by the direction and strength of the horizontal oceanic currents that transport heat, characterized by the barotropic stream function. The barotropic stream function refers to the circulation of ocean currents at a certain depth, where the flow is primarily influenced by pressure gradients. Changes in the barotropic stream function can cause shifts in the paths of ocean currents. As a result, the direction and strength of heat transport in the ocean may be affected. This, in turn, leads to changes in the distribution of ocean heat content across different regions. Thus, the barotropic stream function provides further knowledge about the the paths of ocean currents."

*L149 Why do we know that the heat is coming from the ocean?*

We thank the reviewer for bringing this up. Since the mechanism we analyzed evolves over several years in the North Atlantic ocean, which includes the accumulation of heat, we conclude that the ocean is warming the atmosphere via the ocean-atmosphere heat flux rather than the atmosphere is cooling the ocean. Our conclusion is also supported by the positive sign of the heat flux anomaly (indicative of heat flux transfer from the ocean to the atmosphere). We added: "Based on this dynamical linkage we conclude that the ocean is warming the atmosphere via the ocean-atmosphere heat flux rather than the atmosphere is cooling the ocean. Our conclusion is also supported by the positive sign of the heat flux anomaly, indicative of heat flux transfer from the ocean to the atmosphere."

*L156 How do you know that this is a block and not just a high-pressure system?*

The fact that we analyze summertime means, together with the weaker jet stream, leading to more stationary weather conditions, lead to the conclusion that this is at least a long-living high pressure system, favoring a blocking situation. A detailed analysis is in fact pending. We have rewritten this sentence just saying that this sea level pressure pattern shows the structure of a Scandinavian Blocking. "5-10 year bandpass-filtered sea level pressure anomalies, reveal a structure of a Scandinavian Blocking, which can be identified considering years with and without extremely warm summers".

*L158 How does it confirm the connection between the sub-decadal variability – please explain in more detail.*

We agree with the editor, that this sentence needs to be expanded to explain the mentioned connection, which refers to the identification of the link from the accumulation of heat in the North Atlantic Ocean over specific atmospheric conditions (such as Scandinavian Blocking) to extremely warm European summers. We extended and rewrote the sentence accordingly: " The Scandinavian Blocking can drive heat extremes over Central Europe (Spensberger et al., 2020), and connects the sub-decadal North Atlantic Ocean heat accumulation leading via specific atmospheric conditions to extremely warm summers over Central Europe"

*L160 The link between weaker jet streams and blocks is still contested, there are also arguments that a weaker temperature gradient results in a reduced blocking frequency see e.g., https:// agupubs.onlinelibrary.wiley.com/doi/full/10.1002/2014GL060764*

We thank the editor for bringing this to our attention, to reflect this, we softened our statement to "Additionally, some studies show that the weakening of wind speeds during extremely warm European summers can increase the probability of atmospheric blocking (Woollings et al., 2018), which would in turn increase the likelihood of heat extremes (Kautz et al., 2022)."

*L161 I disagree with the statement that you show how the heat fluxes lead to blocking, this point needs to be further substantiated.*

We have to apologize for this confusion. Our statement is perhaps a bit too overstated and we plan to tone down our wording. Here, we want to say that the heat flux anomalies can influence via the warming the atmosphere the atmospheric circulation and could thus affect the occurrence of long-lasting high-pressure systems such as blocking. Change to " Here, we showed that long-term North Atlantic Ocean heat inertia and associated accumulation of heat lead to an above average ocean-atmosphere heat flux, which in turn can influence the atmospheric circulation and could further affect the occurrence of long-lasting high-pressure systems, prescribing blocking".

*Figure 1 Please indicate how many events contribute to the composites*

We have to apologize for the confusion. In figure 1 we actually do not show composites. Figure 1a and figure 1b show the dominant time scales of European surface air temperature variability in MPI-GE and ERA5 by using a cross-spectral analysis, figure 1c shows the amount of extremely warm European summers on sub-decadal time scales, and figure 1d the power spectrum of Central European surface air temperature. For our analysis we can investigate 557 extremely warm European summers in MPI-GE, we added this number to the corresponding text.

*Figure 1c: I do not understand the unit*

We must apologize for this confusion. This refers to the number of extreme summers that occur during a simultaneous positive sub-decadal phase in European temperatures per grid cell. The

number of extremely warm European summers (T>90th percentile) per ensemble member refers to the entire time period (73 years). This means that from the approximately 1 in 10th summers that are extreme, as exceeding the 90th temperature percentile, as many as shown in the figure occur during a positive phase of the different regions. We can see that on average over all 100 members more extreme summers   linked to sub-decadal (5-10 year) variations occur over Central Europe than in other regions. We agree with the editor that the unit of this figure is not completely clear, therefore, we revised the figure, now indicating the percentage of extremely warm European summers occurring in positive sub-decadal phases per grid point, such as Figure A.

[Figure]

**Figure A:** New figure 1c, percentage of all heat extremes (T>90th percentile) occurring in a positive bandpass filtered phase (T_bandpass>0) per grid-point in MPI-GE. The blue box defines the region of interest for further analysis (Central Europe, ~15°-35°E; 45°-60°N).

*Figure 2: dots are very hard to see*

We agree that the dots are hard to see in a print-out. We increased the dot size in order to increase their visibility.

**Response to the reviewer #1:**

We highly appreciate and are very thankful for the time and effort that was invested in reviewing our manuscript. The detailed and constructive feedback helped us to improve the manuscript. In the following, we provide an answer to each comment brought up by the editor and reviewers. The original comments are in italic red while our responses are in black.

*In this work, the mechanism of sub-decadal climate variability leading to extreme summer temperature events over Europe are investigated by using a Large Ensemble (LE) performed with the MPI-ESM model. The results are relevant in the context of climate prediction, the analyses are robust and well presented, as the quality of the figures is good. However, the language used in this manuscript is not precise and rigorous enough for a scientific article. There are many inaccuracies that make the text difficult to understand. Surprisingly the second part of the article, from section 3.3 is much better written than the first part.*
*Please, revise your manuscript and improve the precision of the language to make the text clearer. Below some examples and other remarks:*

We thank the reviewer for the comments and for sharing our view on relevance of this paper in the context of climate prediction. We would like to apologize for the confusion that has been caused by non-precise wording. In the following, we will go into more detail on how we intend to eliminate these and other sources of ambiguity.

*1.Abstract. The abstract is a very important part in the article. It needs to be well written. Please consider to rewrite the abstract to be more precise. Below an example of a rewritten abstract. This precision should be used throughout the text.*

*The internal variability of European summer temperatures has been linked to various mechanisms from sub-seasonal to multi-decadal timescales. However, the mechanisms controlling sub-decadal (< 10 years) variations remain unexplored. We find that sub-decadal time scales (3-5 years) dominate summer temperature variability over large parts of the European continent. We show that extremely warm summers over Europe, occurring in sub-decadal periods, are related by a strengthening of the Atlantic Ocean subtropical gyre, an increase of meridional heat transport, and an accumulation of ocean heat content over the North Atlantic several years prior to the extreme event episode. The ocean warming affects the ocean-atmosphere heat fluxes, leading to a weakening and northward displacement of the jet stream and increased probability of occurrence of atmospheric blockings over Scandinavia. Thus, our findings link the occurrence of extremely warm European summers to the thermal inertia of the North Atlantic Ocean, whose potential to improve the predictability of extremely warm summers several years ahead is of great societal interest, especially in a warming climate.*

We would like to express our appreciation for this very detailed suggestion on how exactly to rewrite the abstract. We revised the abstract as following: "The internal variability of European summer temperatures has been linked to various mechanisms on seasonal to sub- and multi-decadal timescales. We find that sub-decadal time scales dominate summer temperature variability over large parts of the continent and determine a mechanisms controlling extremely warm summers on sub-decadal time scales. We show that the sub-decadal warm phases of bandpass-filtered European summer temperatures, hereinafter referred to as extremely warm European summers, are related to a strengthening of the North Atlantic ocean subtropical gyre, an increase of meridional heat transport, and an accumulation of ocean heat content in the North Atlantic several years prior to the extreme summer. This ocean warming affects the ocean-atmosphere heat fluxes, leading to a weakening and northward displacement of the jet stream and increased probability of occurrence of high pressure systems over Scandinavia. Thus, our findings link the occurrence of extremely warm European summers to the accumulation of heat in the North Atlantic Ocean, and provide the potential to improve the predictability of extremely warm summers several years ahead which is of great societal interest".

1. *Line 16: "increase in variability" needs to be more precise.*

   By "increase in variability" we refer to the increase in internal temperature variability, which, together with the general temperature increase, leads to more extreme European summers, as demonstrated by the cited studies. To clarify this, we use changed "increase in variability" to "increase in internal temperature variability".

2. *Line 21: Remove "as part of the large-scale…."*

   We removed "as part of the large-scale….".

3. *Line 25: "In particular" means what here ???*

   "In particular" refers to "long memory mechanisms" at the beginning of the sentence. However, we agree, that this wording could be clearer, therefore we splited the sentence into two parts and mentioned „long-memory mechanisms" again in the second sentence. We changed " … variabilty, in detail such as…" to "…variability. This means in detail long term mechanisms such as…".

4. *Different ocean-related quantities: Define*

   The different ocean-related quantities refer e.g. to the overturning stream function, ocean heat content, barotropic stream function, and sea surface temperature. In order to specify this term, we changed this sentence to "). The variability in the North Atlantic region has been shown to include a fully coupled atmosphere-ocean cycle with a period of about 7-10 years shown for different ocean-related quantities, such as ocean heat content and barotropic stream function".

5. *Line 28: Please, include all the references at the end of the sentence and not in the middle. This is applicable to the rest of the article.*

   We agree with the reviewer that references at the end of sentences could enhance the readability and thus contribute to the clarity of the text. We changed this in the revised manuscript whenever it seemed appropriate to us.

6. *Lines 31-33: Please rewrite, not very understandable.*

   We have rewritten this section to "This research aims to address these inquiries and present a comprehensive explanation for the occurrence of unusually hot European summers attributing it to the heat accumulation that occurs several years in advance. Our investigation concentrates on the exceptionally warm European summers that occur in conjunction with positive sub-decadal periods."

7. *In the introduction there should be a reference to DCPP-C (AMV pace makers experiments) and their findings. I think that the findings of this project are relevant and need to be mentioned in this article.*

   Thanks for this suggestion. We agree with the reviewer and added a reference to the Decadal Climate Prediction Project (DCPP) and in particular their AMV pace maker experiments when mentioning the Atlantic multi-decadal variability, such as Boer et al. (2016), Ruprich-Robert et al. (2021), or Qasmi et al. (2021).

   Boer, G. J., Smith, D. M., Cassou, C., Doblas-Reyes, F., Danabasoglu, G., Kirtman, B., Kushnir, Y., Kimoto, M., Meehl, G. A., Msadek, R., Mueller, W. A., Taylor, K. E., Zwiers, F., Rixen, M., Ruprich-Robert, Y., and Eade, R.: The Decadal Climate Prediction Project (DCPP) contribution to CMIP6, Geosci. Model Dev., 9, 3751–3777, https://doi.org/10.5194/gmd-9-3751-2016, 2016.

   Qasmi, S., Sanchez-Gomez, E., Ruprich-Robert, Y., Bo., J., and Cassou, C.: Modulation of the Occurrence of Heatwaves over the Euro-Mediterranean Region by the Intensity of the Atlantic Multidecadal Variability, Journal of Climate, 34, 1099–1114, https://doi.org/10.1175/jcli-d-19-0982.1, 2021.

   Ruprich-Robert, Y., Moreno-Chamarro, E., Levine, X. *et al.* Impacts of Atlantic multidecadal variability on the tropical Pacific: a multi-model study. *npj Clim Atmos Sci* **4**, 33 (2021). https://doi.org/10.1038/s41612-021-00188-5

8.  *The term "Multi-year" is not precise. It is better to use sub-decadal.*

    We agree and changed "multi-year" to "sub-decadal" in the revised manuscript.

9.  *We use the largest ensemble…are you sure ?? I think that other LE also provide 100 members.*

    Sorry for this confusion. To our knowledge there is one additional 100-member large ensemble available with historical and future scenario simulations from a fully coupled climate model, the CESM2-LE. Although this ensemble consists oft wo sets of 50-members under slightly different forcing conditions and provides only one future scenario, we changed our wording to "one of the largest" to avoid ambiguities.

10. *What's a single model initial condition large ensemble?*

    Single-model initial condition large ensemble refers to a term widely used in literature to describe a set of simulations from a single climate model under the same forcing conditions but starting from different initial conditions. We decided to change the sentence to avoid this ambiguity: "MPI-GE offers one of the most adequate representations of observed historical temperatures among single-model large climate models."

11. *North Atlantic ocean heat inertia, ocean is missing several times in the text after North Atlantic*

    We thank the reviewer for this comment and added "ocean" throughout the text.

12. *Remove "doing so", it's too familiar*

    We removed "doing so" throughout the text.

13. *Line 41: We confirm that the MPI-GE can represent sub-decadal temperature variability well…, where?*

    This result can be found in section 3.1, where we analyze whether MPI-GE exhibits good representation of sub-decadal time scales in terms of their dominant time frequency with a cross-spectral analysis (fig. 1a compared to fig. 1b). and check how many extremely warm European summers can generally be found per grid point (fig. 1c).

14. *Remove" by identifying periods that increase….summers"*

    We removed this part of the sentence.

15. *…are done -> are performed*

    We changed "are done" to "are performed".

16. *12-150 km à ??? what does it mean ?*

    We revised and expanded the section on the ocean model grid/resolution. The MPI-M ocean model employed in our study (MPI-OM in GR15 resolution, Marsland et al. 2003) is formulated on a C grid and orthogonal curvilinear coordinates. To circumvent grid singularities at the geographical North Pole, the northern grid pole is shifted to Greenland, leading to high resolution in the Arctic and the high-latitude sinking regions. The horizontal resolution is about 1.5° on average and varies from a minimum of 12km close to Greenland to a maximum of 180km in the tropical Pacific. The model has 40 vertical non-equidistant z levels, of which 20 are distributed in the upper 700m. The coupled model does not employ flux adjustment.
    We changed this section to "In the ocean, the MPI-ESM-LR is formulated on a C grid and orthogonal curvilinear coordinates (Marsland et al., 2003). To circumvent grid singularities at the geographical North Pole, the northern grid pole is shifted to Greenland, leading to high resolution in the Arctic and the high-latitude sinking regions. In the ocean, the MPI-ESM-LR has 40 vertical levels and a horizontal resolution of about 1.5° on average and varies from a minimum of 12km close to Greenland to a maximum of 180km in the tropical Pacific."

*Marsland, S. J., Haak, H., Jungclaus, J. H., Latif, M., & Röske, F. (2003). The Max Planck Institute global ocean/sea ice model with orthogonal curvilinear coordinates. Ocean Modelling, **5**, 91– 127.*

17. *ERA5 reanalysis -> include reference*

We already included a reference at the end of the sentence (see Hersbach et al., 2018 line 64).

18. *MPI-GE can represent the sub-decadal time scales well -> with respect to ???*

We analyze whether MPI-GE exhibits good representation of sub-decadal time scales in terms of their dominant time frequency compared to ERA5. This means whether the model accurately captures the frequency at which significant variations and patterns occur within periods of 5-10 years compared to ERA5. We added "ERA5" in the revised manuscript.

19. *Please rewrite sentence Doing so, ….*

For more clarity and in order to avoid "doing so, …", we combined this sentence with the previous sentence. "We perform the multitapering for all 100 ensemble members and take the mean over all spectra for each grid point to ascertain the dominant timescale, where the dominant timescale is given by the highest significant pea [Arthun et al., 2018, Ghil et al., 2002]."

20. *Remove "around the red noise"*

We removed "around the red noise".

21. *Are you using daily data ? how the summer is defined here ?*

We are using monthly data averaged to seasonal summer means over June, July, and August. We added the temporal resolution, as well as the study domain to the method section.

22. *The way the temperature extremes are defined in MPI-GE should be explained in the "Methods" section.*

We agree and added the following definition "In order to investigate extremely warm European summers on sub-decadal timescales, hereafter referred to as extremely warm summers, we consider those JJA mean temperature anomalies in the region between 15°-35°E and 45°-60°N exceed their 90th percentile and additionally occur in a positive bandpass-filtered phase (pooled in time and ensemble (T>90th percentile and Tbandpass >0), 557 summers in total)" to the method section.

23. *I understand that the computations are done separately for each member in MPI-GE ? So what it is presented in the figures ? the ensemble mean of the quantity ?*

As the reviewer already mentioned we perform the spectral analysis individually for each ensemble member in MPI-GE. Afterwards we take the mean over all spectra for each grid point. For every grid point at least one peak was significant after a chi-squared 95% interval, if more than one peak for a specific grid point were significant, the more highly significant one was chosen. We hope that this answers the question, we added the fact that we take the mean over all spectra afterwards to the multi-taper method section.

24. *Line 81. This is the conclusion of the paper, this is indeed what you want to prove…*

We agree that this statement might be misleading in this paragraph. We moved this sentence to the conclusions.

25. *Which variable is used as "temperature" to determine the extremes? surface temperature, 2m temperature ?*

For all of our analyses we used the temperature at 2m height (T2m). We added this to the revised manuscript.

26. *Line 87. In summary, sub-decadal timescales of 5-10….i do not understand the link to MEAN summer temperature since the focus are here the extremes…*

Here, we used "mean" since the cross spectral analysis is performed for the whole time series, not only extremes (a homogenous time series is needed for this kind of analysis). In the first figure only sub-figure 1c refers to extremely warm European summers, which are defined as those with mean summer temperatures above the 90th percentile. We have to apologize for this confusion, we revised figure 1c and added more text to explain the connection between the dominant time scales (fig. 1b) and percentage of extremely warm summers occurring in positive sub-decadal phases (fig. 1c).

27. *Lines 90-91, Figure 1.d. I see that there are significant peaks at periods of 15 and 2-4 years. I would not conclude that the sub-decadal is the dominant frequency here…*

We agree with the reviewer that the peak at about 15 years is the most dominant one, however, with this figure we want to show that the sub-decadal time scales also have significant peaks over Central Europe and thus we have a good reason to analyze them further. We added: "Other significant peaks could be found around two to three years and around 15 years, indicating the possible influence of other drivers and mechanisms."

28. *Line 94. North Atlantic ocean heat content variability*

We thank the reviewer for this comment and added "ocean" and "content".

29. *Figure1.a and 1.b are not performed at the same spatial resolution, however it is mentioned that ERA5 is regridded on MPI grid, can you clarify ?*

We have to apologize for this confusion. We decided to maintain the figure as it is without regridded data and removed this erroneous statement from the manuscript.

30. *Line98. Please be more accurate and define "years" throughout the text.*

We agree, that we haven't used "years" in an appropriate way in this manuscript. We plan to use "…up to three years prior to an extreme summer" instead of "…years prior to an extreme summer". We changed this throughout the manuscript.

31. *These high "positive" anomalies…*

We added "positive" to this sentence.

32. *Line 104. Some relation to other long-term climate variability models over the Pacific…can you specify??*

In order to be more concrete, we added the Pacific Decadal Oscillation and Tripolar Pacific Index as examples for possible other modes of ocean variability to this sentence: "Although the global anomaly pattern suggests some relation to other long-term climate variability modes over the Pacific, such as the Pacific Decadal Oscillation and Tripolar Pacific Index …".

33. *Please, rewrite lines 106-110.*

We changed these sentences to "This means that no specific ENSO phase (El-Nino, La-Nina, Neutral) can be concretely associated with extremely warm European summers on sub-decadal time scales. Whether this relationship is coincidental and caused by an extraneous process (Cane et al., 2017), or whether this response is indicating a dynamical relationship between

processes in the North Atlantic Ocean and the occurrence of extremely warm European summers, is further investigated in the following."

34. *In addition to ENSO, have you consider the TPI (Tripolar Pacific Index) or IPV ? Anomalies shown in fig.2 look very much to TPI spatial structure.*

Thanks for this comment, first we only tested a possible teleconnection with ENSO as one of the most well-known and influential teleconnections for European climate. However, of course it is worth mentioning that the ENSO teleconnection is just one of many factors that can influence European weather patterns, and other climate phenomena, such as the TPI, might also play a role. Therefore we also tested the influence of TPI on our analysis can came to a similar conclusion as for the ENSO teleconnection.

35. *Line 120. Please, define the meridional ocean heat transport better.*

We have included in the revised manuscript a citation of the meridional ocean heat transport computation and mentioned that the different parts are calculated independently, none of them is a residual. Changed to "Here, further insight into the dynamics of the North Atlantic Ocean subtropical and subpolar region is provided by the 5-10 year bandpass-filtered ocean heat transport and its decomposition into a gyre- and meridional circulation part (Fig. 3b; calculated independently, see Ghosh et al. (2023)."

36. *Line 145. "During the extreme." Please be more accurate.*

"During the extreme" refers to the year when the extremely warm European summer occurs, which is lag 0. We added "… at lag 0".

37. *149. "temperature anomalies fit…" do you mean spatially fit ??*

Indeed we mean "spatially fit" here. We added this to the corresponding sentence.

38. *Line 150. The heat transfer from the ocean to the atmosphere.*

We have rewritten this sentence as suggested by the reviewer: "The transfer of heat from the ocean to the atmosphere is strong enough that its signal reaches up to 200 hPa altitude, with a peak in the range of 400-600 hPa".

39. *Line 152. Extremely European warm summers.*

As suggested by the reviewer, we use "extremely warm European summers" instead of "heat extremes".

40. *Line 165: The North Atlantic OCEAN heat inertia…*

Thanks again for this comment. As already written, we changed "North Atlantic" to " North Atlantic Ocean" throughout the text.

41. *Line 168. "together with an above average…" of what??*

"Together with an above average" refers to the volume transport of the barotropic stream function mentioned later in the sentence. We can see where this confusion comes from and have rewritten the sentence to make it clearer to what this refers to: "Positive anomalies of the ocean heat content, together with an above average horizontal volume transport as well as more northern horizontal volume transport of the barotropic stream function lead to a stronger North Atlantic current and accumulation of heat content. "

42. *Lines 170-171. Please, specify where the "released heat" goes…to the atmosphere??*

Indeed the reviewer is right and the released heat goes into the atmosphere. We added "to the atmosphere" in the corresponding sentence.

43. *Line 181. Rewrite: We find that the coupled oscillation in the North Atlantic prescribes extremely warm European summers on sub-decadal timescales.*

We changed this sentence to "We find that the coupled oscillation in the North Atlantic influences the occurrence of very hot summers in Europe on sub-decadal time scales."

44. *Line 182. Other modes of ocean variability? Could you specify?*

In order to be more concrete, we added the North Atlantic Oscillation, El Niño-Southern Oscillation, and Pacific Decadal Oscillation as examples for possible other modes of ocean variability to this sentence: "However, on this timescale other modes of oceanic variability, such as the North Atlantic Oscillation, El Niño-Southern Oscillation, or Pacific Decadal Oscillation may also be influential."

45. *There are other mechanisms leading to the occurrence of extreme events over Europe-Mediterranean, this should be mentioned in the text. Please see Qasmi et al. 2021 and references therein.*

We agree that extreme events are often the result of a combination of multiple factors and that also other mechanisms could lead to the occurrence of extreme events over Europe. We added a paragraph discussing the influence of the NAO and AMV: "Additionally, our findings might also be impacted by other mechanisms that interact with each other and possibly lead to the occurrence of extreme events over Europe. For example the North Atlantic Oscillation (NAO) and the Atlantic multi-decadal variability (AMV) exert significant influence on the occurrence of extreme events over Europe (Scaife et al., 2008; Qasmi et al., 2021). The NAO plays a crucial role in shaping weather patterns, contributing to the development of heatwaves and droughts, while variations in the AMV can impact atmospheric circulation patterns, influencing the frequency and intensity of extreme events."

46. *The fact that the study was conducted with only one climate model should be discussed in the last section, as the mechanism described in this study could change from one model to another… so there is still uncertainty linked to the model used. We are not sure that a different Large Ensemble performed with a different model would give the same results.*

We agree with the reviewer and added the following paragraph "This study is a single model study which allows us to delve deeper into specific processes and model intricacies, which can contribute to model improvement and process understanding. Replicating this analysis for different climate models would be of great importance to sample potential model uncertainty in these results and help us gain further understanding of this mechanism."

**Response to the reviewer #2:**

We highly appreciate and are very thankful for the time and effort that was invested in reviewing our manuscript. The detailed and constructive feedback helped us to improve the manuscript. In the following, we provide an answer to each comment brought up by the editor and reviewers. The original comments are in italic red while our responses are in black.

*General Remarks*
*The study covers a lot of ground, connecting 5-10 year European summer temperature variability to ocean heat content variability, and posits a plausible and interesting relationship between the two in the MPI-GE. There are sections where the writing is clear and sections that could use some work (e.g. with formality, dangling comparisons). My main concern with the study is the lack of context given along the way from a lead-lag relationship between climatic fields to the causality implied in the title "Extremely Warm European Summers driven by Sub-Decadal North Atlantic Heat Inertia" and throughout. The results are based on a single climate model, run with relatively low resolution, in a region that has some notoriously "obstinate" SST biases likely tied to unresolved oceanic processes (e.g. Athanasiadis et al. 2022). In addition, the North Atlantic sector has been identified as a region where the ocean-atmosphere coupling is weaker in models than in observations, creating issues for NAO variability, blocking, and decadal prediction (e.g. Simpson et al. 2018; albeit focused on winter, but that is the season the coupling is stronger to begin with). For the conclusions made in this paper to stand as they are written, there will need to be a convincing argument made for each link in the causal chain that the MPI-GE lack significant biases with respect to observed fields and that X indeed induces Y. For example, the "driver" is concluded to be a build-up of heat in the North Atlantic Current, but that accumulation ultimately can be traced back to atmospheric variability, right? If you were to impose the ocean heat anomaly in the model, would the atmosphere respond as you describe? I would recommend toning down the conclusions to reflect what is really being explored, the relationship between North Atlantic OHC variability and periods of exceptionally warm summers in the MPI-GE.*

> Thank you for your detailed and constructive review of the study. We appreciate your insightful comments and concerns regarding the context, limitations, and uncertainties of our findings. We acknowledge that our study relies on a single climate model with relatively coarse resolution, which can introduce biases and affect the representation of oceanic processes. We agree that a comprehensive assessment of causality requires further investigation and a convincing argument for each link in the causal chain, including the influence of atmospheric variability on the build-up of heat in the North Atlantic current. We carefully considered these suggestions and provided a more nuanced interpretation of our results, focusing on the relationship between North Atlantic ocean heat content variability and exceptionally warm summer. As an example we have toned down the conclusion and the title of the manuscript, which is now "Extremely Warm European Summers preceded by Sub-Decadal North Atlantic Heat Accumulation".

*Specific Comments*
*L3: Re: "... remain unexplored": Some work appears to be done in the realm, including by the co-authors (e.g. Müller et al. 2020).*

> We agree with the reviewer that this sentence is misleading. Indeed we cite several other studies later in the introduction section. Therefore changed "…and the mechanisms controlling such sub-decadal variations remain unexplored." to "…and determine a mechanisms controlling extremely warm summers on sub-decadal time scales."

*L4-5: Please revise this sentence for clarity.*

> For more clarity we splited this sentence into two parts: ". We show that extremely warm summers over Europe, occurring in sub-decadal periods, are related by a strengthening of the North Atlantic ocean subtropical gyre, an increase of meridional heat transport, and an accumulation of ocean heat content in the North Atlantic several years prior the extreme

event episode. The ocean warming affects the ocean-atmosphere heat fluxes, leading to a weakening and northward displacement of the jet stream and increased probability of occurrence of atmospheric blockings over Scandinavia".

*L19: Consider highlighting the recent work of Röthlisberger and Papritz (2023).*

We thank the reviewer for this comment and cited Röthlisberger and Papritz (2023).

*L29-31: I'm not convinced this is true. I've included a few potential references, but I feel a deeper dive into the literature is warranted.*

We agree with the reviewer that the statement of this sentence is perhaps a bit too overstated and therefore toned down and reworded this sentence: "However, the assessment of drivers for extreme temperatures on long-term timescales is currently limited (Simpson et al., 2018, Wu et al., 2019), and their relevance for extreme summers remains uncertain (Röthlisberger et al., 2023)."

*Simpson, I. R., Deser, C., McKinnon, K. A., & Barnes, E. A. (2018). Modeled and Observed Multidecadal Variability in the North Atlantic Jet Stream and Its Connection to Sea Surface Temperatures. Journal of Climate, 31(20), 8313–8338. https://www.jstor.org/stable/26508075*

*Röthlisberger, M., Papritz, L. Quantifying the physical processes leading to atmospheric hot extremes at a global scale. Nat. Geosci. 16, 210–216 (2023). https://doi.org/10.1038/s41561-023-01126-1*

*Wu, B., Zhou, T., Li, C. et al. Improved decadal prediction of Northern-Hemisphere summer land temperature. Clim Dyn 53, 1357–1369 (2019). https://doi.org/10.1007/s00382-019-04658-8*

*L36-37: Best in what way?*

"One of the best" means one of the most adequate representations of observed historical temperatures among the climate model large ensembles available at the time of the study. We specified this: "MPI-GE offers one of the most adequate representations of observed historical temperatures among single-model large climate models available at the time of the study (Suarez-Gutierrez et al., 2021).

*L39: Re: "Including some of the most extreme European summers": What do you mean by this? Extreme compared to what?*

"Most extreme" means one of of the most exceptional European summer temperatures ever recorded. We added "…including some of the most extreme European summer temperatures ever recorded" this for clarification.

*Section 2.1 Model Description: Maybe in this section, you could also note your study domain and the fields you will use for each part on the analysis*

Thanks for this suggestion. We agree with the reviewer and added the temporal resolution, as well as the study domain to the method section: "Our research focuses on seasonal summer means (JJA) over Central Europe, defined as an area of 15°-35°E/45°-65°N as well as the whole North Atlantic Ocean area."

*L55: Parenthetical should be its own sentence.*

We thank the reviewer for this suggestion and changed from the parenthetical to an own sentence.

*Section 2.2: This section is structured in a very atypical way. Please revise and avoid the use of sub-headings.*

We agree with the reviewer and rewrote this section as a continuous text without sub-headings.

*L63: How precisely do you determine the significance of spectral peaks?*

To determine the significance of spectral peaks in the presence of red noise spectrum with a 95% confidence interval, we compare the amplitude of the peak to a threshold value derived from the red noise spectrum. If the amplitude of the peak exceeds the threshold, it is considered statistically significant at the 95% confidence level.

*L65: Is this a linear detrending? Is that appropriate for "all of [y]our data"?*

In our case, we have chosen a linear detrending to allow comparisons to ERA5. Both linear detrending and removing external forcings by subtracting the ensemble mean yield similar results in this case, as shown in Fig. B. Especially, for the heat extremes as peaks of the time series there is no difference in the timing of their occurrence. In contrast to other results (e.g. Borchert et al., 2021), a linear detrending in the MPI Grand Ensemble seems appropriate and does not distort the results. This difference may be due to the model type of model used here, an un-initialized fully coupled Earth-System-Model. We clarified in the revised manuscript which detrending method we used.

[Figure]

**Figure B:** Comparison between detrending methods in MPI-GE. Exemplary the Central European mean summer temperature time series for both detrending methods, linear detrended and detrended by subtracting the ensemble mean are displayed. Here, for simplicity, only one ensemble member and one variable are displayed, however the authors statement is also true in a broader context.

*L69: Randomly composed arrays of..?*

Here „randomly composed arrays of the corresponding variable" is missing and is added to the revised manuscript.

*L74/76: I'm a bit lost. What is meant by: "total summer mean variability during extremely warm summers"? Is this interannual variability? Decadal? Assessed only during warm periods? How are "for times when heat extremes occur" defined?*

The "total summer mean variability during extremely warm summers" is given by the standard deviation of unfiltered summer (JJA) mean anomalies (calculated with respect to their long-term averages) for years when extremely warm European summers. This means it is the inter-annual summer variability assessed for warm periods. We agree that our phrasing appears to be unnecessary complicated and therefore we have rewritten this section: "…we scale the band-pass filtered summer mean anomalies by   the standard deviation of unfiltered summer (JJA) mean anomalies during extremely warm European summers, calculated with respect to their long-term averages."

"For times when heat extremes occur" refers to years that match with our definition for extremely warm summers, to make this clear, we added a definition of extremely warm European summers to the method section.

*L79-80: "Also, timescales between 10 and 20 years are dominant in only a few more grid points compared to timescales above 20 years." What do you mean by this?*

We refer in this sentence to the number of significant grid points found for the different time intervals in the cross-spectral analysis and want to express that for the 10-20 year interval only few grid points are found, but for the interval above 20 years even fewer are found and here reanalysis and GE are not consistent. We apologize for the confusion and revised this sentence: "On time scales between 10 and 20 years, only a few grid points are dominant. Even fewer dominant grid points are found on time scales greater than 20 years."

*Figure 1:*

• *I thought the ERA5 grid was decimated to match the MPI-GE grid?*

We have to apologize for this confusion. We decided to maintain the figure as it is without regridded data and removed this erroneous statement from the manuscript.

• *There seems to be disagreement on the dominant timescale of SAT variability in your study region between ERA5 and the MPI-GE. Could you comment on that?*

Indeed the dominant time scales in the reanalysis and the model disagree on the broader region of sub-decadal dominance. However, assuming that certain real-world processes may be simulated by climate models correctly albeit for the wrong regions, we find the agreement between the the model and the reanalysis very striking. Although the model simulates the dominance of sub-decadal timescales for temperature in a wider and slight more eastward region, it still captures its effect. Therefore, the model can still be useful to understand this mechanism and its drivers, accounting for the biases in the region of influence. We have expanded our discussion section to elaborate on this issue. Our mechanism still has great relevance for the real world, even if in a somewhat deviated/shifted region. The results from Müller et al. (2020) as well as first results of our current ongoing research confirm the validity of our statements to the real world.

*Müller, W. A., Borchert, L., & Ghosh, R. (2020). Observed Subdecadal Variations of European Summer Temperatures. doi: 10.1029/2019gl086043*

• *There are 6 extremely warm summers per 5–10-year period in each ensemble member? How can (almost) every summer be extreme?*

We must apologize for this confusion. The number of extremely warm European summers per ensemble member refers to the entire time period (73 years) not to 5-10 year intervals. This means that not approximately every summer is extreme, but only every 10th summer. We can see that on average over all 100 members more extreme summers occur over Central Europe than in other regions. We agree with the reviewer that this figure is perhaps not completely clear, this confusion is in our opinion mainly due to the unclear description of the unit. Therefore, we revised the figure, now indicating now the percentage of extremely warm European summers occurring in positive sub-decadal phases per grid point, such as Figure C.

[Figure]

**Figure C:** New figure 1c, percentage of all heat extremes (T>90th percentile) occurring in a positive bandpass filtered phase (T_bandpass>0) per grid-point in MPI-GE. The blue box defines the region of interest for further analysis (Central Europe, ~15°-35°E; 45°-60°N).

• *It may make the figure too messy, but it would be nice to see the power spectra of each individual member, maybe in a supplement? And isn't the dominant variability cycle at around 15 years?*

We agree with the reviewer that the peak at about 15 years is the most dominant one, however, with this figure we want to show that the sub-decadal time scales also have significant peaks over Central Europe and thus we have a good reason to analyze them further. We added a sentence discussing the peaks on other time scales. Further, we agree that it would be helpful to see the spectra of the individual ensemble member, however the single spectra of our 100 ensemble members are basically all over the place and thus provide no additional knowledge and would require a larger y-axis range. We decided to leave the figure as it is in order to make it not too messy and to focus with the chosen y-axis range on the ensemble mean spectrum.

*Section 3.3: What's missing here is validation that the low-resolution ocean model can capture these processes.*

In fact a validation analysis for MPI-GE is pending. However, there are indications that the processes are valid in forced-ocean experiments (such as those in Müller et al. (2020), using the same ocean model in their figure S1) and other coupled climate models (e.g. Martin et al. (2019). These results underline the importance of ocean heat content accumulation for summer mean climate, and their relation to a damped sub-decadal oscillation behavior in the coupled North Atlantic. However, a detailed analysis  is beyond the scope of this study but of importance for further research.

*L116-117: What initiates this? How is it related to the AMO?*

We thank this reviewer for this question. For this study we analyzed which mechanisms in the North Atlantic Ocean drives extremely warm European summers up to three years prior their occurrence. Indeed the influence of external drivers leading to this anomalies within the North Atlantic Ocean is pretty interesting, but to include a thorough analysis accounting for more modes of variability we find beyond the scope of the paper.

*L149-150: How do you know this is "bottom-up" driven and not "top-down" driven?*

Since the mechanism we analyzed evolves over several years in the North Atlantic ocean, which includes the accumulation of heat, we conclude that the ocean is warming the atmosphere via the ocean-atmosphere heat flux rather than the atmosphere is cooling the ocean. Our conclusion is also supported by the positive sign of the heat flux anomaly (indicative of heat flux transfer from the ocean to the atmosphere). We added: "Based on this dynamical linkage we conclude that the ocean is warming the atmosphere via the ocean-atmosphere heat flux rather than the atmosphere is cooling the ocean. Our conclusion is also supported by the positive sign of the heat flux anomaly, indicative of heat flux transfer from the ocean to the atmosphere".

*Figure 4: Are these the ensemble means?*

Yes, the jet-stream position is calculated by the ensemble mean once for members showing and extremely warm European summer and once for members showing no extremely warm European summer. We added "…the mean position of the jet stream averaged over years showing an extremely warm European summer".

*Figure 5: Including the OHC branch of the mechanism in the schematic would be helpful.*

We have to apologize for the confusion. In figure 5, the pink pluses should already illustrate the increasing ocean heat content and the accumulation of heat. We added a figure caption which explains the single elements of the figure, which make the figure hopefully easier to understand.

*Citations to consider*

*Athanasiadis, P. J., and Coauthors, 2022: Mitigating Climate Biases in the Midlatitude North Atlantic by Increasing Model Resolution: SST Gradients and Their Relation to Blocking and the Jet. J. Climate, 35, 6985–7006, https://doi.org/10.1175/JCLI-D-21-0515.1.*

*Simpson, I. R., Deser, C., McKinnon, K. A., & Barnes, E. A. (2018). Modeled and Observed Multidecadal Variability in the North Atlantic Jet Stream and Its Connection to Sea Surface Temperatures. Journal of Climate, 31(20), 8313–8338. https://www.jstor.org/stable/26508075*

*Röthlisberger, M., Papritz, L. Quantifying the physical processes leading to atmospheric hot extremes at a global scale. Nat. Geosci. 16, 210–216 (2023). https://doi.org/10.1038/s41561-023-01126-1*

*Hall, R.J., Jones, J.M., Hanna, E. et al. Drivers and potential predictability of summer time North Atlantic polar front jet variability. Clim Dyn **48**, 3869–3887 (2017). https://doi.org/10.1007/s00382-016-3307-0*

*Osborne, J. M., M. Collins, J. A. Screen, S. I. Thomson, and N. Dunstone, 2020: The North Atlantic as a Driver of Summer Atmospheric Circulation. J. Climate, 33, 7335–7351, https://doi.org/10.1175/JCLI-D-19-0423.1.*

*Wu, B., Zhou, T., Li, C. et al. Improved decadal prediction of Northern-Hemisphere summer land temperature. Clim Dyn 53, 1357–1369 (2019). https://doi.org/10.1007/s00382-019-04658-8*

We want to thank the reviewer for providing these citations. Indeed we find them very helpful and implemented some of them in the revised version of this manuscript.

---

## Author Response (AR2)

**Response to the reviewer #1**

We highly appreciate and are very thankful for the time and effort that was invested in reviewing our manuscript. The detailed and constructive feedback helped us again to improve the manuscript. In the following, we provide an answer to each comment brought up by the reviewer. The original comments are in italic red while our responses are in black.

**1. Comments to revised manuscript:**

*The paper has improved, but some context is still lacking in the introduction; a more comprehensive overview of the literature would help the reader to better understand the novelty of the findings.*
*In terms of content, the analysis of the ocean segment of the proposed mechanism is interesting and clearly described, but the mean temperature section is still difficult to follow. Why focus on a region with a clear variability bias between MPI and ERA5? How robust is the ensemble mean spectrum compared to a pooled spectrum? Isn't the logic a bit circular that a 5-10-year period with many warm extremes is a positive sub-decadal phase?*

We thank the reviewer for his/her feedback and would like to address a few points here in general before the specific responses follow afterwards: We have carefully extended the introduction section, now mentioning the coupled ocean-atmosphere cycle and added some appropriate citations. Further, we have revised mean temperature section, reworded the corresponding sections and changed/added some figures, we hope that these changes contribute to the understanding.

*Specific Comments:*
*L33-34: I'm not sure how this follows from the paper... Please see note on the attached document.*

Thanks for bringing this up. This comment is answered in detail below in section 2 "Comment of previous reviewer #2 referring to L19" and "Comment of previous reviewer #2 referring to L29-31".

*L47-48: Maybe soften "responsible for" to "associated with" or "precede". Additionally, ...which processes ... are...*

We agree with the reviewer and changed "Additionally, we identify which processes in the North Atlantic Ocean is responsible for the increase of the occurrence of extremely warm summers." to "Additionally, we identify which processes in the North Atlantic Ocean precede the increase of the occurrence of extremely warm summers."

*L59: we*

Changed.

*L59: What variables are you using?*

We analyzed anomalies of the ocean heat content, the ocean heat transport, the barotropic stream function, and the ocean-atmosphere heat flux, as well as anomalies of the vertical temperature and the temperature at surface. We added these variables to to corresponding sentence and changed the sentence from "Here, we are using monthly data averaged to seasonal summer means over June, July, and August (JJA) from 1950 to 2022." to "Here, we are using monthly data averaged to seasonal summer means over June, July, and August (JJA) from 1950 to 2022 and analyzed anomalies of the ocean heat content, the ocean heat transport, the barotropic stream function, and the ocean-atmosphere heat flux, as well as anomalies of the vertical temperature and the temperature at surface."

*L61: Incomplete sentence*

We have to apologize, we could not find any incomplete sentence in the mentioned lines. However, we added a missing comma: "For time-lagged analyses, up to three years prior to 1950 are analyzed".

*Section 2.2 Please revise this section into paragraphs. Rarely should one sentence be set off as its own paragraph in scientific writing.*

Thank you for this comment. We have revised this section in a way that individual sentences no longer form a separate paragraph.

*There is a break in line numbers for some reason, so the following are based to the 60:*
*L60+4: linearly*

We changed "linear" to "linearly".

*L60+8: temperature as in 2-m temperature (tas)?*

We have to apologize for this imprecise wording. We changed "...we consider those JJA mean temperature anomalies..." to "...we consider those JJA mean 2m air temperature anomalies..."

*L60+11-12: What do you mean by "if and where...can represent the sub-decadal timescales"? Timescales of what? By definition, isn't ERA5 the target? Why would it not be able to represent the sub-decadal timescales?*

Our goal here is to investigate how MPI-GE can represent subdecadal temperature variability compared to ERA5. This investigation is intended to test the assumption that real-world processes are correctly represented in climate models (such as the MPI-GE here). For a better understanding, we changed the sentence from "We use a cross-spectral analysis, based on a multi-taper method to analyze if and where the MPI-GE and ERA5 can represent the sub-decadal time scales." to "We use a cross-spectral analysis, based on a multi-taper method to analyze how the MPI-GE can represent the sub-decadal temperature variability compared to ERA5."

*L60+23-25: Can you show the following as regional average timeseries, even just for a single ensemble member?*
*- unfiltered JJA mean anomalies, with extremely warm summers marked*
*- bandpass filtered JJA mean anomalies*

Yes, we can show such a figure. The figure illustrates that not necessarily all warm summers occur in a positive bandpass-filtered phase and show that summers which occur in a positive phase are clustered together with other extreme years.

[Figure]

**Figure A.** Extremely warm European summers on sub-decadal time scales (for illustration purposes only ensembles member 50 is shown). Detrended time series of Central European surface air temperature anomalies, non-filtered (blue) and 5-10 year bandpass-filtered (orange). Heat extremes (above 90th percentile) that occur in a positive bandpass-filtered phase (therefore fulfill our selection criterium) are marked in green, while heat extremes that occur in a negative bandpass-filtered phase are marked in red.

*Line numbering resumes:*
*L65-66: I don't understand the phrase: "the proportion ... has on the..."*

To clarify this, we changed the sentence from "The scaled anomaly simply illustrates the proportion that a sub-decadal mean change has on the occurrence of an extremely warm summers compared to the overall occurrence of extremely warm summers."
to "The scaled anomaly simply illustrates the impact of sub-decadal processes on the occurrence of an extremely warm summers compared to the overall occurrence of extremely warm summers."

*L73-74: But they disagree over your study region? The dominant mode in ERA5 is below 5 years, while MPI-GE shows a dominant 5-10 year mode, right? This is why I recommended decimating your ERA5 grid to the MPI-GE grid; it's not "fair" to compare a low-resolution model's "spatially average" climate to a high-resolution reanalysis and make statements about regional behavior.*

We unterstand the doubts raised by the reviewer and regridded the EAR5 data therefore to the coarser MPI-GE grid. As pointed out by the reviewer, there are differences in the key region of this study. We now have added some further explantations in the manuscript that reflects the reasoning of the regional differences between the model and ERA5. However, we think that the model is able to capture the principle large-scale distribution of the sub-decadal variations compared to ERA5, thereby making it suitable to investigate the large-scale drivers. In the subsequent analysis, we concentrate on the model world to establish the mechanism. We added:"The cross-spectral analysis reveals, that MPI-GE is able to capture the large-scale distribution of the dominant sub-decadal variations compared to ERA5. This points towards the ability of the model to simulate the underlying large-scale mechanism in principle. However, still there are regional differences in the distribution, as can be found for example over South-East Europe. The reasons are yet unclear, and can be related to a regional displacement of the principal modes of large-scale variability compared to ERA5, or the limitation of the model to reflect the impact of the regional sea on extremes in this region (Beobide et al., 2023)."

[Figure]

**Figure B.** Dominant time frequencies and their relation to extremely warm European summers. (a),(b) Cross-spectral analysis, performed using the multi-taper method, showing the dominant time scales of European surface air temperature variability in ERA5 (a) revised figure and (b) figure 1a of the main manuscript.

*L78-79: Doesn't the presence of heat extremes in a particular 5-10 year period then determine that temperature is in a positive phase?*

Thanks for this comment, we have to admit that this sentence is not very precise. Although extremes occurring within the 5-10 year period is not a necessary criteria, these positive phases occur over periods clustered warmer than normal, eventually extreme, summers. This means it is neither a sufficient nor a required criteria, but there may be a correlation between the two. A non-filtered extreme needs not necessarily to occur in a positive bandpass-filtered phase (see Fig. A). To clarify this we changed "Analyzing the ratio between all heat extremes and those occurring in a positive bandpass filtered phase, Central Europe stands out as the area with the highest percentages." to "Analyzing the ratio between all non-filtered heat extremes and those occurring in a positive bandpass filtered phase, Central Europe stands out as the area with the highest percentages.

*Section 3.2: Please revise the one-sentence paragraphs*

We have also revised this section to avoid individual sentences forming a separate paragraph.

*L117-118: Is this shown in the figure?*

The value is calculated as given in section 2.2, offering a possible way to interpret the values given in Fig. 3a. However, since this value is already given in the figure caption, we decided to remove this sentence from the main text.

*L157: Average in time or across members or both?*

We want to thank the reviewer for bringing this up. We are actually using the average in time and across ensemble members during extremely warm European summers. We changed the sentence from "In addition, the average position of the jet stream is shifted northward..." to "In addition, the average position of the jet stream, in time and across ensemble members, is shifted northward...".

*L170: in the MPI-GE.*

We agree with the reviewer and changed "The North Atlantic Ocean heat accumulation impacts the occurrence of extremely warm summers over Central Europe on sub-decadal timescales." to "The North Atlantic Ocean heat accumulation impacts the occurrence of extremely warm summers over Central Europe on sub-decadal timescales in MPI-GE."

*L177-186: This seems like it should be in the introduction rather than only in the discussion.*

We thank the reviewer and extended the introduction section. "... Further, the variability in the North Atlantic region has been shown to include a fully coupled atmosphere-ocean cycle with a period of about 7-10 years shown for various atmosphere- and ocean-related quantities, such as sea surface temperature and Gulf Stream indices (Czaja et al., 2001; McCarthy et al., 2018), ocean heat content and overturning stream functions (Martin et al., 2019), prominent winter sea-level pressure patterns (Czaja et al., 2001), and the North Atlantic Oscillation (DaCosta et al., 2002). This cycle is associated with an active role of the atmospheric heat and momentum forcing, together with a delayed effect of the redistribution of North Atlantic water masses (Czaja et al., 2001; Eden and Greatbach, 2003; Reintges et al., 2016; Martin et al., 2019). In fact, this process have a significant impact on European summer temperatures as demonstrated by Müller et al., 2020). However, the assessment of drivers for extreme temperatures on such long-term timescales is currently limited (Simpson et al., 2018; Wu et al., 2019), and their relevance for extreme summers remains uncertain."

*L213-214: Please revise.*

We agree and changed the sentence from "Lastly, this is a single model study which allows us to delve deeper into specific processes and model intricacies, which can contribute to model improvement and process understanding" to "Lastly, replicating this analysis for different climate models would be of great importance to sample potential model uncertainty in these results and help us gain further understanding of this mechanism".

*L217-221: These sentences are repeats of the previous paragraph.*

We agree and removed the doubled sentences.

*L233-234: Is this a reemergence signal? If so, doesn't that connect it to European summer climate*

The reviewer is right, as also stated in the text, the described mechanism can be seen to be attached to a fully coupled atmosphere-ocean cycle evolving in a 7-10 year period. Such oscillating behavior shows indeed a reemergence signal. We added this detail to the text: "Analyzing longer lags, in this case lag -7 to 0, prior to extremely warm European summers shows that the described mechanism can be seen as attached to a reemerging fully coupled atmosphere-ocean cycle evolving in a 7-10 year period."

**2. Comments to answers to previous reviewer #2:**

*Comment of previous reviewer #2 referring to L19. Please be sure to read the paper carefully and cite it appropriately. "the main drivers of extreme heat are soil moisture deficits and moisture-temperature feedbacks" - This is not what the study found over Europe; heat in your study region is primarily adiabatic compression and subsidence.*

Original comment of former reviewer #2 and four answer: *L19: Consider highlighting the recent work of Röthlisberger and Papritz (2023).*

We thank the reviewer for this comment and cited Röthlisberger and Papritz (2023).

We thank the reviewer for this clarification, which we address together with the following comment below.

*Comment of previous reviewer #2 referring to L29-31. Again I'm not sure if this is a true statement based on the reference. They assess heat on shorter timescales and mention uncertainty in how the drivers will evolve given future circulation (inherently uncertain).*

Original comment of former reviewer #2: *L29-31: I'm not convinced this is true. I've included a few potential references, but I feel a deeper dive into the literature is warranted.*

We agree with the reviewer that the statement of this sentence is perhaps a bit too overstated and therefore toned down and reworded this sentence: "However, the assessment of drivers for extreme temperatures on long-term timescales is currently limited (Simpson et al., 2018, Wu et al., 2019), and their relevance for extreme summers remains uncertain (Röthlisberger et al., 2023)."

*Simpson, I. R., Deser, C., McKinnon, K. A., & Barnes, E. A. (2018). Modeled and Observed Multidecadal Variability in the North Atlantic Jet Stream and Its Connection to Sea Surface Temperatures. Journal of Climate, 31(20), 8313–8338. https:// www.jstor.org/stable/ 26508075*

*Röthlisberger, M., Papritz, L. Quantifying the physical processes leading to atmospheric hot extremes at a global scale. Nat. Geosci. 16, 210–216 (2023). https://doi.org/10.1038/s41561-023-01126-1*

*Wu, B., Zhou, T., Li, C. et al. Improved decadal prediction of Northern-Hemisphere summer land temperature. Clim Dyn 53, 1357–1369 (2019). https://doi.org/10.1007/s00382-019-04658-8*

We thank the reviewer for these two comments and cite Röthlisberger and Papritz (2023) now in a more appropriate way. For line 19 (now line 20) we decided to move the citation of Röthlisberger and Papritz (2023): "On time scales of days to several weeks, the main drivers of extreme heat are soil moisture deficits and moisture-temperature feedbacks (Seneviratne et al., 2006; Fischer and Schär, 2008; Vogel et al., 2017; Suarez-Gutierrez et al., 2020a), diabatic heating, adiabatic compression and advection (Röthlisberger and Papritz, 2023), and large-scale atmospheric patterns such as atmospheric blocking and the North Atlantic Oscillation (Meehl and Tebaldi, 2004; Horton et al., 2015; Li et al., 2020; Suarez-Gutierrez et al., 2020a). However, these short-term drivers of extreme temperatures could be influenced and conditioned by mechanisms on longer time scales". For line 32-34 we decided to remove the citation.

*Comment of previous reviewer #2 referring to Section 2.1. What variables are you using?*

Original comment of former reviewer #2: *Section 2.1 Model Description: Maybe in this section, you could also note your study domain and the fields you will use for each part on the analysis*

Thanks for this suggestion. We agree with the reviewer and added the temporal resolution, as well as the study domain to the method section: "Our research focuses on seasonal summer means (JJA) over Central Europe, defined as an area of 15°-35°E/45°-65°N as well as the whole North Atlantic Ocean area."

We analyzed anomalies of the ocean heat content, the ocean heat transport, the barotropic stream function, and the ocean-atmosphere heat flux, as well as anomalies of the vertical temperature and the temperature at surface. We added these variables to to corresponding sentence and changed the sentence from "Here, we are using monthly data averaged to seasonal summer means over June, July, and August (JJA) from 1950 to 2022." to "Here, we are using monthly data averaged to seasonal summer means over June, July, and August (JJA) from 1950 to 2022 and analyzed anomalies of the ocean heat content, the ocean heat transport, the barotropic stream function, and the ocean-atmosphere heat flux, as well as anomalies of the vertical temperature and the temperature at surface."

*Comment of previous reviewer #2 referring to L65. What is being shown here? There is no y-axis label.*

Original comment of former reviewer #2: *L65: Is this a linear detrending? Is that appropriate for "all of [y]our data"?*

In our case, we have chosen a linear detrending to allow comparisons to ERA5. Both linear detrending and removing external forcings by subtracting the ensemble mean yield similar results in this case, as shown in Fig. B. Especially, for the heat extremes as peaks of the time series there is no difference in the timing of their occurrence. In contrast to other results (e.g. Borchert et al., 2021), a linear detrending in the MPI Grand Ensemble seems appropriate and does not distort the results. This difference may be due to the model type of model used here, an un-initialized fully coupled Earth-System-Model. We clarified in the revised manuscript which detrending method we used.

We have to apologize for this confusion. The y-axis of the figure is showing the temperature anomalies [°C].

*Comment of previous reviewer #2 referring to Figure 1. Could you comment on this in the manuscript?*

Original comment of former reviewer #2: *There seems to be disagreement on the dominant timescale of SAT variability in your study region between ERA5 and the MPI-GE. Could you comment on that?*

Indeed the dominant time scales in the reanalysis and the model disagree on the broader region of sub-decadal dominance. However, assuming that certain real-world processes may be simulated by climate models correctly albeit for the wrong regions, we find the agreement between the the model and the reanalysis very striking. Although the model simulates the dominance of sub-decadal timescales for temperature in a wider and slight more eastward region, it still captures its effect. Therefore, the model can still be useful to understand this mechanism and its drivers, accounting for the biases in the region of influence. We have expanded our discussion section to elaborate on this issue. Our mechanism still has great relevance for the real world, even if in a somewhat deviated/shifted region. The results from Müller et al. (2020) as well as first results of our current ongoing research confirm the validity of our statements to the real world.

Müller, W. A., Borchert, L., & Ghosh, R. (2020). Observed Subdecadal Variations of European Summer Temperatures. doi: 10.1029/2019gl086043

We unterstand the doubts raised by the reviewer and regridded the EAR5 data therefore to the coarser MPI-GE grid. As pointed out by the reviewer, there are differences in the key region of this study. We now have added some further explantations in the manuscript that reflects the reasoning of the regional differences between the model and ERA5. However, we think that the model is able to capture the principle large-scale distribution of the sub-decadal variations compared to ERA5, thereby making it suitable to investigate the large-scale drivers. In the subsequent analysis, we concentrate on the model world to establish the mechanism. We added:"The cross-spectral analysis reveals, that MPI-GE is able to capture the large-scale distribution of the dominant sub-decadal variations compared to ERA5. This points towards the ability of the model to simulate the underlying large-scale mechanism in principle. However, still there are regional differences in the distribution, as can be found for example over South-East Europe. The reasons are yet unclear, and can be related to a regional displacement of the principal modes of large-scale variability compared to ERA5, or the limitation of the model to reflect the impact of the regional sea on extremes in this region (Beobide et al., 2023)."

*Comment of previous reviewer #2 referring to Figure 1. In this case, it would be interesting to see what the spectrum would be if you pooled all the ensemble members and computed the spectrum.*

Original comment of former reviewer #2: *It may make the figure too messy, but it would be nice to see the power spectra of each individual member, maybe in a supplement? And isn't the dominant variability cycle at around 15 years?*

We agree with the reviewer that the peak at about 15 years is the most dominant one, however, with this figure we want to show that the sub-decadal time scales also have significant peaks over Central Europe and thus we have a good reason to analyze them further. We added a sentence discussing the peaks on other time scales. Further, we agree that it would be helpful to see the spectra of the individual ensemble member, however the single spectra of our 100 ensemble members are basically all over the place and thus provide no additional knowledge and would require a larger y-axis range. We decided to leave the figure as it is in order to make it not too messy and to focus with the chosen y-axis range on the ensemble mean spectrum.

We thank the reviewer for this comment and added the requested pooled ensemble spectrum below. However, we do not plan to add this spectrum to the manuscript, since the pooled ensemble features discontinuities at the points where individual members are concatenated, which leads to artificially induced errors in the resulting spectrum. Therefore, we plan to continue with the mean of all ensemble member spectra.

[Figure]

**Figure C.** Power spectrum of Central European surface air temperature (black line) in MPI-GE (pooled ensemble spectrum). The significance is shown via a red-noise spectrum (solid red line) and the chi-squared 95% interval (dashed red line). Period 1950-2022. (For comparison, see Fig. 1d main manuscript.)

---

## Author Response (AR3)

**Response to the editor:**

We highly appreciate and are very thankful for the time and effort that was invested in reviewing our manuscript. The original comments are in italic red while our responses are in black.

*Please change the wording in Figure 5 to not imply causation at least for the causal arguments that pertain to the atmosphere and the jet.*

We want to thank the editor for this comment. We changed the corresponding sentences from "… an increased ocean-atmosphere heat flux. The released heat causes then an above average warming of the atmosphere reaching even high altitudes. This warming of the atmosphere leads via jet stream displacement to extremely warm European summers." to "… an increased ocean-atmosphere heat flux. The released heat is linked to an above average warming of the atmosphere reaching even high altitudes. This warming of the atmosphere is connected to a jet stream displacement and extremely warm European summers."

[Figure]

**Fig I:** Revised version of figure 5 of the main manuscript.